# HiDrop: Hierarchical Vision Token Reduction in MLLMs via Late Injection, Concave Pyramid Pruning, and Early Exit

**Hao Wu**[1,2*]  **Yingqi Fan**[1*]  **Jinyang Dai**[3]  **Junlong Tong**[1,2,4]  **Yunpu Ma**[5]  **Xiaoyu Shen**[1,2†]

[1]Institute of Digital Twin, Eastern Institute of Technology, Ningbo
[2]Ningbo Key Laboratory of Spatial Intelligence and Digital Derivative
[3]University of Science and Technology of China    [4]Shanghai Jiao Tong University
[5]Munich Center for Machine Learning, LMU Munich
`haowu.ai.research@gmail.com`    `xyshen@eitech.edu.cn`

## Abstract

The quadratic computational cost of processing vision tokens in Multimodal Large Language Models (MLLMs) hinders their widespread adoption. While progressive vision token pruning offers a promising solution, current methods misinterpret shallow layer functions and use rigid schedules, which fail to unlock the full efficiency potential. To address these issues, we propose HiDrop, a framework that aligns token pruning with the true hierarchical function of MLLM layers. HiDrop features two key innovations: (1) Late Injection, which bypasses passive shallow layers to introduce visual tokens exactly where active fusion begins; and (2) Concave Pyramid Pruning with an Early Exit mechanism to dynamically adjust pruning rates across middle and deep layers. This process is optimized via an inter-layer similarity measure and a differentiable top-$k$ operator. To ensure practical efficiency, HiDrop further incorporates persistent positional encoding, FlashAttention-compatible token selection, and parallel decoupling of vision computation to eliminate hidden overhead associated with dynamic token reduction. Extensive experiments show that HiDrop compresses $\sim$90% visual tokens while matching the original performance and accelerating training by $1.72\times$. Our work not only sets a new state-of-the-art for efficient MLLM training and inference but also provides valuable insights into the hierarchical nature of multimodal fusion. The code is released at https://github.com/EIT-NLP/HiDrop.

## 1 Introduction

Multimodal Large Language Models (MLLMs) have driven rapid progress in fields ranging from visual question answering to embodied AI by seamlessly bridging vision and language (OpenAI, 2023; 2024; Lu et al., 2024; Bai et al., 2025; Achiam et al., 2023). Most modern MLLMs rely on a connector-based architecture (Liu et al., 2023a;b; 2024a; Bai et al., 2023; Wang et al., 2024), wherein a lightweight module projects visual features directly into the LLM's embedding space, allowing a pre-trained text backbone to process multimodal inputs without retraining from scratch (Li et al., 2023a). However, visual encoders typically generate substantially more tokens than text due to their higher information density (Dosovitskiy et al., 2020). With token count scaling quadratically with image resolution and self-attention quadratic in token number, computation quickly becomes prohibitive (Lin et al., 2024; Cui et al., 2024).

To address this issue, a common strategy involves progressive vision token pruning (Chen et al., 2024b; Xing et al., 2024; Yao et al., 2025), where less informative visual tokens are gradually eliminated as they move through the model. The underlying intuition is that early layers should maintain dense visual representations to preserve fine-grained details, whereas deeper layers can function effectively with a reduced token set concentrated on semantically significant content (Fan et al., 2025).

---

*Equal contribution.
†Corresponding author.

Figure 1: Comparison of progressive vision token pruning methods. (a) FastV conducts single-stage pruning at an early layer. (b) TwigVLM performs early pruning and removes remaining vision tokens at deeper layers. (c) PDrop applies progressive pruning with uniform ratios and intervals. (d) HiDrop introduces vision tokens only at the end of shallow layers, prunes them in a non-uniform progressive manner in middle layers, and removes remaining vision tokens before deep layers. (e) HiDrop prunes vision tokens by about $4.8\times$ more aggressively than state-of-the-art progressive pruning method with negligible performance drop.

However, after a closer examination of the internal dynamics of such models, we identify that existing pruning approaches are constrained by two core misunderstandings regarding how multimodal large language models (MLLMs) process visual information across different layers.

First, *shallow layers are misinterpreted*. Prior work observes that removing early layers degrades performance and thus concludes that these layers are critical for multimodal integration (Xing et al., 2024; Zhang et al., 2025; Wu et al., 2025). Our analysis shows otherwise: vision tokens, already deeply processed by the vision encoder, undergo almost no transformation in the initial LLM layers. Both intra-modal evolution and cross-modal influence are negligible. These layers primarily act as propagators and attention sinks, not true integrators (Fan et al., 2025).

Second, *pruning schedules are overly rigid*. Existing approaches often adopt fixed-ratio, pyramid-like schemes such as FastV (Chen et al., 2024b), TwigVLM (Shao et al., 2025), and PDrop (Xing et al., 2024). However, we find that visual information flow is highly non-uniform: redundancy can be removed more aggressively in middle layers where fusion dominates, while visual tokens can be safely discarded altogether in the deep layers once integration is complete. Uniform schedules miss this structure and thus lead to suboptimal efficiency–accuracy trade-offs.

Motivated by these findings, we propose **HiDrop**, a hierarchical vision token dropping framework that coordinates token pruning and computation skipping in accordance with the hierarchical processing dynamics of MLLMs across different layers.

To address the shallow-layer misconception, we adopt a Late Injection strategy: rather than pruning in shallow layers, we bypass them altogether and inject the full set of vision tokens only at the onset of the true fusion stage. This approach perfectly reflects the functional redundancy of the early layers without prematurely discarding potentially valuable information, marking the first attempt to deliberately delay, rather than simply prune, visual input for greater efficiency in MLLMs.

To address the limitations of rigid schedules, we propose a *Concave Pyramid Pruning* scheme, which accelerates token reduction early in the fusion stage and slows it later, together with an *Early Exit* mechanism that fully discards vision tokens before the language-dominant layers. When applying this schedule, we identify reliable pruning layers using an *Inter-Layer Visual Attention Similarity (ILVAS)* measure, and select the most informative tokens with a *differentiable top-$k$ operator* (Liu et al., 2024b). These mechanisms jointly enable precise and adaptive pruning decisions.

Beyond algorithmic design, HiDrop employs persistent position identifiers to preserve positional consistency under dynamic token activation, decouples token selection from the main attention computation to remain compatible with efficient kernels such as FlashAttention, and exploits late injection to parallelize vision-related computation with text-only prefill. These designs ensure that dynamic token management does not incur hidden overheads and translate the theoretical efficiency gains into real-world acceleration.

Extensive experiments on LLaVA-1.5-7B show that HiDrop compresses $\sim$90% of visual tokens while matching the original performance, accelerating training by up to $1.72\times$ and substantially improving inference throughput. Our contributions are threefold: (1) we diagnose two fundamental weaknesses of existing pruning methods related to shallow-layer interpretation and pruning sched-

Figure 2: **Layer-wise representational dynamics**, with the left panel showing intra-modal refinement, and the right panel highlighting cross-modal interaction intensity.

ules; (2) we introduce HiDrop, featuring the novel Late Injection strategy, Concave Pyramid Pruning with Early Exit, and optimized layer- and token-selection mechanisms; and (3) we empirically demonstrate that HiDrop achieves state-of-the-art efficiency–accuracy trade-offs.

## 2   UNMASKING THE PROCESSING DYNAMICS IN MLLMS

A Multimodal Large Language Model (MLLM) processes a unified sequence of text and vision embeddings, $\mathbf{h}_0 = [\mathbf{E}_v : \mathbf{E}_t]$, through its Transformer layers. The text embeddings $\mathbf{E}_t \in \mathbb{R}^{N_t \times d}$ come from a standard tokenizer, while the vision embeddings $\mathbf{E}_v \in \mathbb{R}^{N_v \times d}$ originate from a vision encoder that partitions an image into $N_v$ patches and projects their features into the LLM's hidden dimension $d$. The primary computational bottleneck in this architecture is self-attention, whose cost scales quadratically with the number of vision tokens, $\mathcal{O}(N_v^2 d)$, as typically $N_v \gg N_t$.

To mitigate this computational burden, a common solution is **progressive token pruning**, which iteratively reduces the number of vision tokens across the model's layers. However, most existing strategies rely on predetermined and static pruning schedules, such as linear or convex decay, applied uniformly across layers without accounting for the distinct processing dynamics at different stages of the model. This raises a critical question: what constitutes an *effective* pruning strategy? We argue that token pruning should be grounded in the model's actual behavior rather than handcrafted heuristics. Achieving this requires a deeper understanding of how MLLMs internally process and integrate visual information. Therefore, we conduct an in-depth analysis of their internal dynamics, revealing that different layers play fundamentally distinct roles in multimodal fusion, which in turn motivates a more principled pruning approach.

**Shallow Layers: Propagators**   A prevalent assumption in progressive pruning is that shallow layers are essential for early cross-modal fusion and must be preserved (Xing et al., 2024; Zhang et al., 2025). To scrutinize this belief, we perform a *training-free* layer-wise probe on LLaVA-v1.5-7B, feeding GQA image–question pairs through the network and recording hidden states at all layers. Our analysis, however, reveals that these layers function not as active integrators but as simple propagators. We demonstrate this by examining their contributions from two perspectives.

First, we analyze intra-modal refinement by measuring how token representations evolve across layers for each modality $\mathcal{M} \in \{\text{System}, \text{Visual}, \text{Textual}\}$. Concretely, we compute the modality-specific cosine similarity ($\mathbf{S}_{\text{intra}}^{\mathcal{M}}$) between the outputs of consecutive layers:

$$\mathbf{S}_{\text{intra}}^{\mathcal{M}} = \frac{1}{N_{\text{sample}}} \sum_{i=1}^{N_{\text{sample}}} \left( \frac{1}{N_{\mathcal{M}}} \sum_{t \in \mathcal{T}_{\mathcal{M}}} \frac{\langle x_{i,t}^l, x_{i,t}^{l+1} \rangle}{\|x_{i,t}^l\|_2 \, \|x_{i,t}^{l+1}\|_2} \right).$$

where $l$ denotes the layer index, $N_{\text{sample}}$ is the number of samples, $\mathcal{T}_{\mathcal{M}}$ is the set of tokens belonging to modality $\mathcal{M}$ with $N_{\mathcal{M}} = |\mathcal{T}_{\mathcal{M}}|$, and $x_{i,t}^l$ is the representation of token $t$ in sample $i$ at layer $l$.

As shown in the left panel of Fig. 2, visual token representations in the shallow layers exhibit remarkably high self-similarity, undergoing only very minor changes across consecutive layers, indicating that the LLM backbone performs negligible processing on them in this stage.

Second, we measure cross-modal influence by how much text embeddings for a fixed instruction change when paired with different images, and define the resulting cross-modal similarity as $\mathbf{S}_{\text{cross}}^{\text{Ins}}$:

$$\mathbf{S}_{\text{cross}}^{\text{Ins}} = \frac{1}{N_{\text{sample}}} \sum_{i=1}^{N_{\text{sample}}} \frac{\langle \mathbf{h}_{i,\text{ins}}^{(l,\text{mis})}, \mathbf{h}_{i,\text{ins}}^{(l,\text{ref})} \rangle}{\|\mathbf{h}_{i,\text{ins}}^{(l,\text{mis})}\|_2 \, \|\mathbf{h}_{i,\text{ins}}^{(l,\text{ref})}\|_2}.$$

where $\mathbf{h}_{i,\text{ins}}^{(l,\text{mis})}$ is the layer-$l$ instruction embedding for sample $i$ paired with a mismatched image, and $\mathbf{h}_{i,\text{ins}}^{(l,\text{ref})}$ is the counterpart paired with a fixed reference image.

Contrary to common belief, the right panel of Fig. 2 shows that, in shallow layers, text embeddings for a fixed instruction are nearly invariant to the accompanying image, indicating that cross-modal influence is still negligible and meaningful fusion has not yet occurred. Combined with the intra-modal analysis above, these results suggest that shallow layers primarily act as passive conduits, simply passing visual information to deeper layers where substantive processing begins.

**Middle Layers: Sparse Fusion Hubs**  In stark contrast to the passive shallow layers, the middle layers emerge as the primary hubs for cross-modal fusion. At this stage, the model actively integrates visual information, causing textual representations to vary significantly in response to visual input (Fig. 2). This fusion, however, is highly sparse: a small subset of key visual tokens grounds the textual embeddings, rendering the vast majority of other visual tokens redundant. This dual characteristic, being both the center of fusion and the peak of redundancy, makes the middle layers the natural bottleneck for multimodal processing.

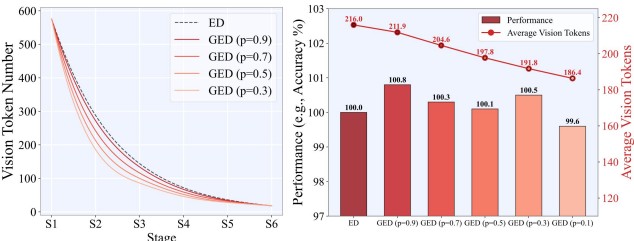

We further substantiate this redundancy with *training-based* pruning experiments. On LLaVA-v1.5-7B, we applied an aggressive middle-layer schedules parameterized by exponential decay (ED) and generalized exponential decay (GED). In GED, an exponent $p$ controls the decay shape, and when $0 < p < 1$ the keep ratio drops much faster in early layers, enabling extremely early pruning. Under an extreme GED schedule that reduces visual tokens from 576 to just 1 across the middle layers, the model still retains 99.6% of its original GQA performance. Moreover, this robustness is not an artifact of a single schedule. As shown in Fig. 3, various alternative pruning strategies also maintain near-perfect accuracy. Such invariance demonstrates that high visual redundancy is a stable, inherent property of the middle layers, making them the ideal location for aggressive token compression.

Figure 3: Left: Vision token reduction curves under different $p$ values, where lower $p$ enforces stronger pruning. Right: Model performance remains stable even under high compression rates, demonstrating robustness of our pruning strategy.

**Deep Layers: Language-Dominant Reasoning**  Once cross-modal fusion is completed in the middle layers, the network transitions into its final stage, which is dominated by abstract, language-centric reasoning. The direct influence of visual tokens steadily diminishes until their role becomes negligible, as seen in Fig. 2. We validate this with behavior on LLaVA-v1.5-7B with a training-free "early exit" experiment, where we discard all visual tokens at a specific layer and observe the impact on performance. As shown in Fig. 4, removing visual tokens in the shallow or middle layers causes a catastrophic performance drop. However, removing them after the main fusion stage (e.g., beyond layer 24) results in almost no degradation. This finding provides strong evidence that the deep layers can operate effectively without direct access to visual information, relying instead on the fused multimodal representations formed in the middle layers. At this point, the network transitions fully into a language-dominant regime to refine semantics and generate the final output.

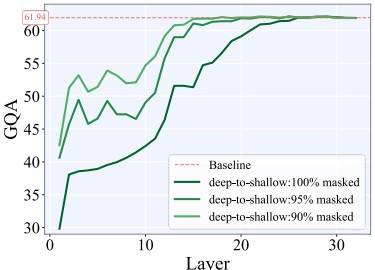

Figure 4: Early vision exit analysis under different masking ratios.

## 3 HIDROP

Building on the insights above, we propose HiDrop (Hierarchical Vision Token Dropping), a framework that adapts pruning to the hierarchical dynamics of MLLMs. As illustrated in Fig. 5, we exploit hierarchical redundancy by partitioning the LLM's layers into shallow, middle, and deep stages: we handle the shallow and deep stages with Late Injection and Early Exit, and apply Concave Pyramid Dropping in the middle stage to progressively reduce vision tokens.

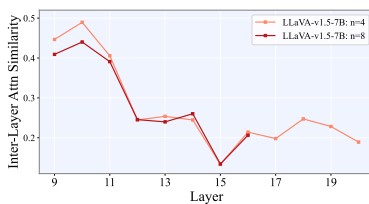

(a) Framework of **HiDrop**

(b) Hard Top-k and Differentiable Top-k.

Figure 5: Overview of **HiDrop**. (a) Framework illustration, shallow layers focus on vision-independent reasoning, middle layers progressively prune redundant tokens through differentiable top-$k$ selection, and deep layers enable early vision exit. (b) Comparison between hard top-$k$ and differentiable top-$k$, which achieves adaptive selection and better information preservation.

## 3.1 SHALLOW AND DEEP: JOINT VISUAL LAYER REDUCTION

As shown in Sec. 2, visual tokens are redundant in both shallow and deep stages. We therefore combine Late Vision Injection, which delays their introduction until fusion begins, with Early Vision Exit, which discards them once language-dominant reasoning takes over.

**Late Vision Injection**  Knowing that shallow layers act as passive conduits (Sec. 2), our approach avoids wasteful computation by employing a *Late Vision Injection* strategy. Instead of processing visual tokens from the first layer, HiDrop bypasses the initial $L_{\text{inj}} - 1$ layers for the visual stream entirely. The text-only forward pass proceeds until the injection layer $L_{\text{inj}}$, where the vision tokens are first introduced and concatenated with the text representations: $\mathbf{h}_{L_{\text{inj}}} = \left[ \mathbf{h}^v_{L_{\text{inj}}} : \mathbf{h}^t_{L_{\text{inj}}} \right]$. This injection point is strategically chosen at the onset of the active fusion stage, which we identify by a local minimum in the visual layer-wise similarity curve (layer 9 in our experiments, Fig. 2).

**Early Vision Exit**  Our analysis in Sec. 2 shows that deep layers transition to a language-dominant regime where direct visual input is no longer required for reasoning. Therefore, HiDrop incorporates an *Early Vision Exit* strategy after a specific exit layer $L_{\text{exit}}$, all remaining vision tokens are discarded, and the forward pass continues with only the text stream. We determine this exit point by identifying where model performance plateaus in our deep-to-shallow masking analysis, indicating that visual tokens are no longer contributing (layer 25, Fig. 4).

Together, Late Injection and Early Exit create a focused "vision processing window," restricting all vision tokens to only middle layers. This targeted approach significantly accelerates both training and inference, all while preserving the model's predictive accuracy.

## 3.2 MIDDLE: AGGRESSIVE CONCAVE PYRAMID PRUNING

Within the core vision processing window, we propose **Concave Pyramid Pruning**, an aggressive yet adaptive strategy to manage the high redundancy found in the middle layers (Sec. 2). This approach is designed to prune tokens rapidly at the start of the fusion stage and then more gradually, preserving essential information while maximizing computational savings. Implementing this strategy requires answering two key questions: (1) *Where* in the middle layers should pruning occur? and (2) *Which* specific tokens should be pruned at these locations?

**Where to Prune: Identifying Filtering Layers with ILVAS** To determine the optimal layers for pruning, we introduce the *Inter-Layer Visual Attention Similarity (ILVAS)* metric. The core idea is to identify layers where the model has formed a stable assessment of token importance, making them ideal 'filtering' points. ILVAS measures how consistently the most attended to visual tokens at one layer remain important in subsequent layers. Specifically, we compare the top-$K$ attention distributions for vision tokens between a layer $l$ and a future

Figure 6: ILVAS curves for different window sizes, extended results in Appendix. G.5.

layer $l + n$:

$$\text{ILVAS}(l, l+n, K) \;=\; \frac{1}{|\mathcal{V}_K^l|} \sum_{i \in \mathcal{V}_K^l} \frac{\left\langle \tilde{\mathbf{A}}_i^l, \; \tilde{\mathbf{A}}_i^{l+n} \right\rangle}{\left\| \tilde{\mathbf{A}}_i^l \right\| \left\| \tilde{\mathbf{A}}_i^{l+n} \right\|},$$

where $\tilde{\mathbf{A}}_i^l$ is the head-wise attention vector for vision token $i$. A high ILVAS score indicates a stable filtering capacity. We compute its curve across the middle layers and select the local maxima to form our set of filtering layers $\mathcal{F}$ (e.g., layers $\{10, 14, 16, 18\}$ in Fig. 6).

**Which Tokens to Prune: Adaptive Selection with Differentiable Top-K**  Once the filtering layers are identified, the next challenge is to select which specific tokens to prune. Previous methods often rely on non-differentiable Hard Top-$K$ selection, which may lead to suboptimal token selection. To overcome this, we employ a Differentiable Top-$K$ (DTop-$K$) operator (Liu et al., 2024b), which provides a continuous relaxation of the selection process..

Given a vector of importance scores $c \in \mathbb{R}^N$ for $N$ tokens, the DTop-$K$ operator first computes a normalized rank score $c'$ for each token: $c_i' = \frac{1}{n} \sum_{j=1}^{n} \mathbb{1}(c_i \geq c_j)$. This maps the scores to a $[0, 1]$ range. Next, a soft mask is generated using a sigmoid function with a learnable pruning ratio $a$:

$$Mask(c, a) = \text{Sigmoid}((c - a) \cdot \lambda) = \frac{1}{1 + e^{-\lambda(c_i' - a)}}.$$

This soft mask produces a smooth importance score for each token, enabling more fine-grained selection than a hard cutoff. For the forward pass, a hard threshold is applied to the mask to make a discrete token selection. By combining ILVAS to determine *where* to prune and DTop-$K$ to determine *which* tokens to prune, our method dynamically and efficiently compresses visual information. A detailed comparison with Hard Top-$K$ is provided in Sec. 4.3.

### 3.3  SOLUTIONS TO IMPLEMENTATION CHALLENGES

**Persistent Position Encoding**  HiDrop dynamically changes which visual tokens are active across layers because of late injection, progressive dropping, and early exit. Naively reindexing tokens under this dynamic behavior can misalign positional encodings. To avoid this, each visual token is assigned a persistent positional identifier at input: although the shallow layers contain no visual tokens, their indices are reserved, activated upon injection, and preserved through subsequent dropping or exit. For RoPE, queries and keys are always rotated using these fixed identifiers, ensuring consistent relative geometry across the model.

**Efficient Attention Compatibility**  To remain compatible with efficient attention kernels such as FlashAttention, the original attention computation is left intact over the full sequence. Token selection is handled separately by a lightweight auxiliary attention pass, restricted to interactions between the final text token and visual tokens. Since this auxiliary step involves only a single query, its overhead is negligible, and the efficiency benefits of HiDrop are fully preserved.

**Parallel Decoupling of Vision-related Operations**  Late injection theoretically allows us to shorten the critical-path prefill time by decoupling vision-related computation from the main attention stack. Before the injection layer, all transformer layers operate purely on text tokens, while in parallel we run the vision encoder once, apply the projector to obtain visual KV tensors, and cache them. At the injection layer, these cached visual KV tensors are concatenated with the text KV tensors, and subsequent layers attend over the combined set. During HiDrop's multi-stage pruning, we only update indices over the cached visual KV tensors instead of recomputing projections. This parallel decoupling removes visual KV projection from the prefill bottleneck and remains compatible with FlashAttention-style kernels.

# 4 EXPERIMENT

## 4.1 EXPERIMENTAL SETTINGS

**Models**  Within the LLaVA-1.5 architecture (Liu et al., 2023a), we verify the effectiveness of the proposed HiDrop with three different LLM backbones: MobileLLaMA-2.7B (Wu et al., 2024), Vicuna-7B-v1.5, and Vicuna-13B-v1.5 (Zheng et al., 2023). The details are provided in Appendix C.

**Benchmarks**  To thoroughly evaluate the HiDrop, we conduct experiments on 11 mainstream benchmarks, including MME$^P$ (Fu et al., 2023), MMB, MMB$^{CN}$ (Liu et al., 2025), GQA (Hudson & Manning, 2019), VQA$^{v2}$ (Goyal et al., 2017), SQA$^I$ (Lu et al., 2022), VizWiz (Gurari et al., 2018), TextVQA (Singh et al., 2019), POPE (Li et al., 2023b), SEED$^I$ (Li et al., 2024a), and MMStar (Chen et al., 2024c). Notably, MMStar (Chen et al., 2024c) is a multimodal benchmark characterized by strong visual dependency and minimal data leakage. See Appendix D for details.

**Efficiency Evaluation**  We consider the efficiency in both training and inference following PDrop (Xing et al., 2024). For training, we report real GPU hours on the same device; for inference, we report FLOPs for vision token part. Specifically, for a Transformer block, the FLOPs from MHA and FFN are $4nd^2 + 2n^2d + 3ndm$, where $n$ is the number of vision tokens, $d$ is the hidden size, and $m$ is the FFN intermediate dimension. Aggregating across layers (with $n_\ell$ denoting the number of vision tokens at layer $\ell$), the total FLOPs are:

$$\text{FLOPs} = \sum_{\ell=1}^{L} \left( 4\,n_\ell d^2 + 2\,n_\ell^2 d + 3\,n_\ell dm \right)$$

**Implementation Details**  For DTop-K operation, we set the temperate $\lambda = N_v$, which means the number of the visual candidate vision tokens. For LLaVA-1.5-7B, we adopt late injection layer $L_{\text{inj}} = 9$, early exit layer $L_{\text{exit}} = 25$, and filtering layers $\mathcal{F} = \{10, 14, 16, 18\}$. For LLaVA-1.5-MobileLLaMA-2.7B, we ues $L_{\text{inj}} = 15$, $L_{\text{exit}} = 28$, and $\mathcal{F} = \{16, 19, 22, 25\}$. All experiments are conducted on 8 NVIDIA A100 40 GB GPUs. Unless otherwise stated, we follow LLaVA's default training (pretrain and instruction finetuning) and evaluation settings for benchmarks included in its suite. The evalution of the MMStar is done via LMMS-Eval (Zhang et al., 2024a) toolkit.

## 4.2 MAIN RESULTS

**Comparison with State-of-the-art Methods**  To ensure a fair comparison, we conduct controlled-budget experiment under three different compression ratio. As shown in Table 1, using LLaVA-1.5-7B as the base LMM, we compare HiDrop against state-of-the-art in-LLM vision token compression methods across eleven widely used benchmarks. HiDrop consistently and markedly outperforms all counterparts at all pruning ratios. Notably, it retains 98.3% and 96.5% of the baseline performance while pruning 88.9% and 91.7% of vision tokens, respectively. Compared with the most similar progressive token pruning approach, PDrop (Xing et al., 2024), HiDrop achieves higher performance on nearly all benchmarks under the 88.9% pruning ratio, with a gap of 4.1% average performance. At even more aggressive compression, HiDrop still retains 96.5% of the baseline at 91.7% pruning, whereas PDrop cannot reach this pruning level under the same protocol.

**Efficiency of HiDrop in Training & Inference**  As shown in Table 2, HiDrop reduces the training time (including both pretraining and finetuning stages) of LLaVA-1.5-7B from 159.3 to 94.4 GPU hours, resulting in an impressive 40.7% reduction in overall time. In addition to the training efficiency improvement, HiDrop also reduces the inference FLOPs from 3.82T to 0.42T, achieving an 88.9% reduction. Moreover, HiDrop lowers the prefill latency from 63.6 ms to 32.6 ms, and can be further reduced to 31.8 ms and 28.8 ms through parallelly decoupled visual KV projection and fewer dropping stages. Notably, compared to PDrop's pruning ratio of 46.9%, HiDrop achieves a much higher pruning ratio of 89.0%, which is 4.8 times more aggressive, while the performance drop is only 1.6%, demonstrating HiDrop's superior efficiency and minimal accuracy trade-off. Similar trends are observed on LLaVA-1.5-MobileLLaMA-2.7B and LLaVA-1.5-13B: across both smaller and larger backbones, HiDrop consistently delivers substantial reductions in training time, FLOPs, and prefill latency under much stronger pruning ratios, while incurring only a slight degradation compared to the vanilla models.

Table 1: Performance comparisons with three pruning ratios on 11 benchmarks. All methods are applied on the same base model **LLaVA-1.5-7B**. The best result for each benchmark and pruning ratio is **bolded**. Dashed lines separate training-free (above) and training-based (below) methods within each block. The $^*$ denotes results reproduced using the official checkpoints; $^\dagger$ denotes training-based methods evaluated under training-free settings; $^\ddagger$ denotes training-free methods evaluated under training-based settings.

| Method | MME$^P$ | MMB | MMB$^{CN}$ | GQA | VQA$^{v2}$ | SQA$^I$ | VizWiz | TextVQA | POPE | SEED$^I$ | MMStar | Avg(%) |
|---|---|---|---|---|---|---|---|---|---|---|---|---|
| *Upper Bound, 576 Tokens (100%)* | | | | | | | | | | | | |
| LLaVA-1.5-7B | 1510.7 | 64.3 | 58.3 | 62.0 | 78.5 | 66.8 | 50.0 | 58.2 | 85.9 | 66.1 | - | - |
| LLaVA-1.5-7B$^*$ | 1506.5 | 64.7 | 58.1 | 61.9 | 78.5 | 69.5 | 50.1 | 58.2 | 86.8 | 66.2 | 33.7 | 100.0 |
| *Retain 80 Tokens in Average (↓ 86.1%)* | | | | | | | | | | | | |
| FastV | 1214.4 | 57.3 | 47.8 | 51.3 | 66.6 | 68.8 | 51.3 | 52.1 | 73.7 | 52.9 | 31.1 | 87.9 |
| PDrop$^\dagger$ | 1133.1 | 53.6 | 41.1 | 50.8 | 67.2 | 69.1 | 46.9 | 51.5 | 72.0 | 50.8 | 30.8 | 84.5 |
| FastV$^\ddagger$ | 1348.2 | 62.3 | 53.1 | 55.4 | 68.9 | 68.8 | 43.4 | 49.1 | 80.6 | 55.3 | 34.7 | 91.3 |
| PDrop | 1412.1 | **64.6** | 54.7 | 57.9 | 74.3 | **69.8** | **52.4** | 54.3 | 83.7 | 59.3 | **35.0** | 96.8 |
| VoCo-LLaMA | 1307.0 | 58.0 | 44.5 | 58.7 | 74.2 | 66.7 | 52.1 | 50.4 | 83.9 | 54.7 | 32.2 | 91.2 |
| TwigVLM | **1471.5** | 62.8 | **56.4** | 59.5 | **76.8** | 69.7 | 51.5 | **56.9** | 85.0 | 60.7 | 34.0 | 97.9 |
| HiDrop (Ours) | 1467.0 | 63.7 | 56.3 | **61.3** | 76.6 | 67.5 | 51.4 | 54.9 | **86.6** | **65.3** | 31.2 | **98.4** |
| *Retain 64 Tokens in Average (↓ 88.9%)* | | | | | | | | | | | | |
| FastV | 1086.6 | 53.3 | 42.7 | 48.8 | 61.6 | 68.9 | 50.5 | 49.9 | 67.7 | 49.1 | 29.6 | 82.8 |
| PDrop$^\dagger$ | 962.0 | 45.6 | 32.7 | 45.4 | 58.3 | 68.2 | 45.9 | 48.2 | 64.0 | 47.3 | 29.0 | 76.5 |
| FastV$^\ddagger$ | 1303.8 | 61.7 | 52.7 | 56.2 | 70.7 | 70.0 | 43.8 | 51.0 | 83.1 | 55.6 | **33.8** | 91.8 |
| PDrop | 1350.7 | 63.1 | 54.3 | 56.6 | 71.8 | **70.3** | 51.8 | 51.7 | 82.6 | 57.9 | 32.7 | 94.2 |
| VoCo-LLaMA | 1256.5 | 55.4 | 44.2 | 58.1 | 73.9 | 66.2 | 51.8 | 49.5 | 83.2 | 54.7 | 33.3 | 90.4 |
| TwigVLM | 1404.0 | 60.4 | 53.6 | 58.8 | 75.6 | 70.0 | 51.2 | **55.8** | 82.7 | 56.9 | 33.1 | 95.3 |
| HiDrop (Ours) | **1473.3** | **63.2** | **58.0** | **60.5** | **76.5** | 68.9 | **52.6** | 55.2 | **86.4** | **64.5** | 32.0 | **98.3** |
| *Retain 48 Tokens in Average (↓ 91.7%)* | | | | | | | | | | | | |
| FastV | 816.9 | 37.3 | 29.8 | 42.1 | 49.6 | 68.7 | 47.6 | 46.3 | 56.1 | 42.4 | 25.6 | 70.2 |
| FastV$^\ddagger$ | 1327.4 | 61.3 | 53.6 | 54.4 | 68.4 | **69.0** | 45.8 | 49.6 | 82.2 | 54.6 | **34.4** | 91.4 |
| VoCo-LLaMA | 1321.9 | 56.2 | 46.2 | 58.6 | 74.1 | 68.1 | **51.8** | 50.9 | 83.9 | 54.9 | 32.3 | 91.6 |
| TwigVLM | 1199.9 | 53.1 | 42.6 | 55.0 | 71.8 | **69.0** | 49.4 | 53.6 | 75.7 | 48.6 | 31.6 | 87.3 |
| HiDrop (Ours) | **1446.4** | **63.7** | **55.5** | **59.8** | **75.6** | 67.7 | 49.5 | **54.4** | **85.8** | **61.8** | 32.7 | **96.5** |

Table 2: Efficiency comparison across three LLM backbones within the LLaVA-1.5 framework. Prefill latency (ms) is reported as actual / decoupled visual-KV / fewer dropping stages.

| Model | Method | Avg. Vis. Tokens | Train hours | Infer TFlops | Prefill Latency (ms) | MME$^P$ | MMB | GQA | VizWiz | VQA$^T$ | Avg(%) |
|---|---|---|---|---|---|---|---|---|---|---|---|
| LLaVA-1.5-MobileLLaMA-2.7B | Vanilla | 576 | 108.4 | 1.52 | 35.3 | 1258.2 | 57.0 | 59.4 | 32.6 | 48.6 | 100.0 |
| | PDrop | 270 | 50.3 | 0.70 | 28.7 | 1231.1 | 54.3 | 57.0 | 30.9 | 47.5 | 96.3 |
| | ours | 64 ↓ 206 | 45.6 | 0.17 | 25.4/25.1/22.0 | 1206.6 | 53.1 | 56.1 | 30.4 | 47.2 | 94.8 ↓ 1.5 |
| LLaVA-1.5-7B | Vanilla | 576 | 159.3 | 3.82 | 63.6 | 1506.5 | 64.7 | 61.9 | 50.1 | 58.2 | 100.0 |
| | PDrop | 270 | 107.3 | 1.78 | 43.7 | 1490.1 | 63.9 | 61.7 | 52.4 | 57.7 | 100.2 |
| | ours | 64 ↓ 206 | 94.4 | 0.42 | 32.6/31.8/28.8 | 1474.3 | 63.2 | 60.5 | 52.6 | 55.2 | 98.6 ↓ 1.6 |
| LLaVA-1.5-13B | Vanilla | 576 | 297.2 | 7.44 | 122.8 | 1529.9 | 68.5 | 63.5 | 53.6 | 61.2 | 100.0 |
| | PDrop | 270 | 213.7 | 3.47 | 74.9 | 1555.2 | 68.8 | 63.1 | 53.7 | 60.8 | 100.3 |
| | ours | 64 ↓ 206 | 175.8 | 0.82 | 48.6/46.6/43.5 | 1497.2 | 66.9 | 62.1 | 56.3 | 58.0 | 98.7 ↓ 1.6 |

## 4.3 ABLATION STUDIES

To better understand the proposed HiDrop, we conduct three group ablation studies to investigate the key attributes of several critical components: (1) Late injection and early exit, assessed independently on the base model; (2) The effect of differentiable top-$k$ and token importance calculation, examined within the progressive dropping setup, where vision tokens are pruned in stages ($576 \rightarrow 64 \rightarrow 8 \rightarrow 1$) at evenly spaced intervals; and (3) Position encoding and filter layer selection, analyzed within the complete shallow-middle-deep compression structure.

Table 3: Performance comparison of LLaVA variants with Hard vs. Differentiable Top-$K$ Operators. PT and FT denote pretrain and finetune, respectively.

| Model | Train | Top-K | MME$^P$ | MMB | GQA | VQA$^{v2}$ | VizWiz | VQA$^T$ | MMStar | Avg(%) |
|---|---|---|---|---|---|---|---|---|---|---|
| LLaVA-1.5-7B | - | - | 1506.5 | 64.7 | 61.9 | 78.5 | 50.1 | 58.2 | 33.7 | 100.0 |
| LLaVA-1.5-7B + Top-$K$ | PT+FT | Hard | 1436.9 | 64.2 | 59.7 | 76.4 | 50.1 | 55.7 | 33.9 | 97.7 |
| | | Diff. | 1484.7 | 65.5 | 60.2 | 76.3 | 52.7 | 56.2 | 34.3 | 99.7 |
| | FT | Hard | 1482.7 | 65.0 | 60.3 | 76.5 | 46.8 | 55.9 | 33.4 | 97.5 |
| | | Diff. | 1471.7 | 65.2 | 59.9 | 76.5 | 47.1 | 56.2 | 34.8 | 98.1 |

Table 4: Effect of different strategies for estimating vision token saliency.

| Model | MME$^P$ | MMB | GQA | VQA$^{v2}$ | VizWiz | VQA$^T$ | MMStar | Avg(%) |
|---|---|---|---|---|---|---|---|---|
| LLaVA-1.5-7B | 1506.5 | 64.7 | 61.9 | 78.5 | 50.1 | 58.2 | 33.7 | 100.0 |
| Last token (1-rounds) | 1424.7 | 65.3 | 59.6 | 75.6 | 49.0 | 55.5 | 33.2 | 97.1 |
| Last token (n-rounds) | 1484.7 | 65.5 | 60.2 | 76.3 | 52.7 | 56.2 | 34.3 | 99.7 |
| Last token (n-rounds, L2 norm) | 1447.0 | 65.2 | 59.7 | 76.3 | 48.8 | 55.9 | 34.0 | 97.9 |
| All token | 1414.8 | 65.0 | 59.0 | 74.8 | 51.4 | 56.6 | 34.3 | 98.1 |
| All token (L2 norm) | 1424.0 | 65.5 | 59.9 | 75.2 | 53.2 | 56.6 | 35.5 | 99.6 |

**Late Injection and Early Exit**  Our late injection and early exit are guided by two diagnostics: layer 9 aligns with a local minimum in the visual layer-wise similarities (Fig. 2), and accuracy plateaus around layer 25 under deep-to-shallow masking (Fig. 4). We validate these choices with three sweeps (Fig. 7). In the late entry sweep, varying the injection layer with the exit fixed shows a clear peak at layer 9; injecting earlier adds cost with little gain, and injecting later degrades accuracy. In the fixed entry span sweep, fixing injection at layer 9 and varying the exit peaks around layers 25 to 26; later exits add cost and earlier exits hurt accuracy. In the equal depth window sweep, sliding

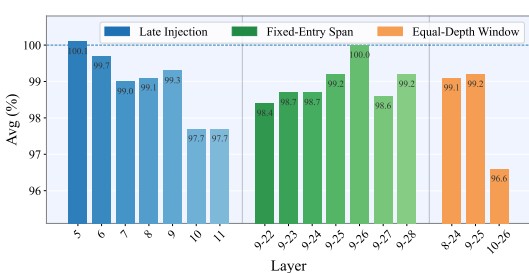

Figure 7: Ablation across visual perception layers comparing *Late Injection*, *Fixed-Entry Span*, and *Equal-Depth Window*, confirming that our setting is the most efficient. The full per-benchmark results are provided in Appendix G.1, Table 8.

a constant-length window confirms 8–24 and 9–25 as near-optimal, while 10–26 underperforms. Notably, in the deep-to-shallow diagnostic, performance matches the baseline at layer 26 and is only slightly lower at layer 25; we therefore choose 25 as the exit, expecting training to recover the small gap, and the sweeps verify that the 9 to 25 window is a strong choice.

**Differentiable Top-K**  We study hard top-$k$ and differentiable top-$k$ under a progressive pruning schedule. As shown in Table 3, replacing hard top-$k$ with differentiable top-$k$ lifts the average performance from 97.7% to 99.7% with two-stage training (pretraining then finetuning) and from 97.5% to 98.1% with one-stage training (finetuning only), indicating more faithful token selection under the same training setting. Since the gain is larger with two-stage training, we adopt this recipe as the default in our experiments. See Appendix G.2 for additional token decay schedules.

**Token Weighting Strategies**  We compare training-time strategies for estimating vision token importance. As shown in Table 4, aggregating attention from all text tokens with L2-norm weighting performs comparably to the multi-round last-token variant, and is in fact 0.3% worse on average across 11 benchmarks (Table 10 in Appendix G.3). Given the extra cost from the eager attention used for importance calculation, we default to the multi-round last-token scheme.

**Position Encoding**  Conceptually, similar to the "position-ID mismatch" in streaming LLMs (Tong et al., 2025b;a; Lin et al., 2026), but distinct in cause: ours arises from cross-layer changes in the set of surviving vision tokens due to late injection (insertion), progressive dropping (pruning), and early exit (removal). We therefore compare three positional encoding (PE) schemes: (1) Persistent PE: assign fixed RoPE indices at input and never update them; (2) Compacted PE (PDrop-style): start with preset indices and, at pruning stages, reset indices to compact surviving vision tokens and

Table 5: Effect of position encoding (PE) schemes under shallow–middle–deep compression.

| Model | MME$^P$ | MMB | GQA | VQA$^{v2}$ | VizWiz | VQA$^T$ | MMStar | Avg(%) |
|---|---|---|---|---|---|---|---|---|
| LLaVA-1.5-7B | 1506.5 | 64.7 | 61.9 | 78.5 | 50.1 | 58.2 | 33.7 | 100.0 |
| Persistent PE | 1414.4 | 63.7 | 61.3 | 76.6 | 52.1 | 55.6 | 32.0 | 97.6 |
| Compacted PE | 1452.3 | 64.6 | 61.1 | 76.8 | 48.9 | 55.1 | 30.3 | 96.4 |
| Group PE | 1442.2 | 63.9 | 60.4 | 76.2 | 51.2 | 55.5 | 31.1 | 97.0 |

Table 6: Effect of instruction fine-tuning data scale (HiDrop retains 48 visual tokens in average).

| Model | Data Scale | MME$^P$ | MMB | GQA | VQA$^{v2}$ | VizWiz$^{val}$ | VQA$^T$ | MMStar | Avg(%) |
|---|---|---|---|---|---|---|---|---|---|
| LLaVA-1.5-7B | 665k | 1506.5 | 64.7 | 61.9 | 78.5 | 54.4 | 58.2 | 33.7 | 100.0 |
| HiDrop | 1M | 1446.4 | 63.7 | 59.8 | 75.6 | 56.3 | 54.4 | 32.7 | 97.0 |
| LLaVA-1.5-7B | 665k | 1526.1 | 68.7 | 62.7 | 79.2 | 61.2 | 58.8 | 38.2 | 100.0 |
| HiDrop | 1M | 1453.9 | 66.2 | 59.5 | 76.1 | 60.7 | 55.4 | 36.9 | 96.3 |

fill gaps; and (3) Group PE: allocate disjoint RoPE index ranges for instruction and vision tokens, with no in-place updates during injection, pruning, or exit. As summarized in Table 5, Persistent PE achieves the best average performance, Group PE is close, and Compacted PE performs worst, consistent with the hypothesis that resetting indices exacerbates cross-layer position mismatch. Given its accuracy and zero overhead, we adopt Persistent PE by default. More benchmark results appear in Appendix G.4.

**Filtering Layer Selection**    We first compute the ILVAS curve over the middle layers on a model configured with late injection and early exit, and select its local maxima as the filtering layers, yielding $\{10, 14, 16, 18\}$ (Fig. 6). To validate this choice, we fix a token–decay schedule that follows the concave pyramid dropping policy and sweep the filtering layers (Fig. 8). Compared with a control schedule $\{12, 15, 18, 21\}$, the ILVAS-based set achieves higher average accuracy. Fixing $\{10, 16, 18\}$ and sweeping the remaining slot produces a clear peak at 14, whereas 12 or 13 degrades performance. Jointly sweeping the middle pair further confirms $\{14, 18\}$ as the best combination; nearby alternatives $\{13, 18\}$, $\{13, 19\}$, and $\{14, 19\}$ underperform. We therefore adopt $\{10, 14, 16, 18\}$ in all main experiments.

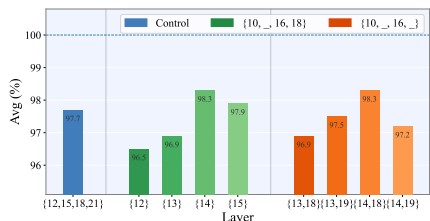

Figure 8: Ablation across filter layers, confirming that our setting is the most efficient. The full per-benchmark results are provided in Appendix G.5, Table 12.

**Training Data Scale**    The HiDrop variant evaluated in Table 6 retains only 48 visual tokens across all settings. We compare the base LLaVA-v1.5-7B and its HiDrop-equipped counterpart under two instruction fine-tuning data scales (665k vs. 1M). As the data scale increases, both the base model and HiDrop consistently improve on most benchmarks (e.g., MMB, MMB-CN, SEED-IMG, MMStar), indicating that HiDrop benefits from additional instruction data rather than being bottlenecked by compression. At the same time, the compressed model remains close to the base model, with average performance drops of only 3.0% (665k) and 3.7% (1M) despite operating under a much more aggressive visual-token budget. These results show that HiDrop tracks the gains of the base model as data scale grows, supporting that our layer-wise compression design is compatible with stronger instruction tuning and that the observed improvements are not artifacts of under-training.

## 5 CONCLUSION

In summary, our study challenges prevailing assumptions about visual processing in MLLMs and demonstrates that shallow layers only act as passive propagators for visual tokens. By introducing HiDivDrop with Late Injection, Concave Pyramid Pruning, and Early Exit, we align pruning with the true hierarchical dynamics of multimodal integration. Our findings not only achieve state-of-the-art efficiency–accuracy trade-offs, but also provide new insights into how MLLMs allocate computation across layers, paving the way for more principled and scalable multimodal architectures.

ETHICS STATEMENT

This work does not present any ethical concerns. Our research focuses on methodological contributions and efficiency analysis without involving sensitive data, human subjects, or applications that could raise ethical risks.

REPRODUCIBILITY STATEMENT

We have taken several steps to ensure the reproducibility of our work. All experimental settings, including dataset descriptions, training details, and hyperparameter selections, are clearly documented in the main text and appendix. We further provide extensive ablation studies to justify our design choices. Upon acceptance, we will release the full codebase and scripts to facilitate replication of our results.

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

## A  THE USE OF LARGE LANGUAGE MODELS

We employed large language models (LLMs) solely as general-purpose writing assistants for language refinement, including improving clarity, grammar, and style. Importantly, no LLM was involved in research ideation, methodological design, analysis, or result interpretation; the role of the LLM was limited to linguistic polishing. All substantive contributions originated from the authors. This ensured that the scientific content remained entirely authored by the researchers, while benefiting from improved academic writing quality.

## B  RELATED WORK

A distinctive property of MLLMs is that vision tokens are far more numerous yet information-sparse compared to text tokens (Marr, 2010), making them the primary source of redundancy and motivating research on token compression. Most prior work is training-free, pruning vision tokens during inference via heuristic rules (Chen et al., 2024b; Zhang et al., 2024b; Yang et al., 2025; Liu et al., 2024c). While effective in reducing computation, these methods introduce a train–inference mismatch. To address this issue, training-based approaches learn token reduction end-to-end, achieving alignment between training and inference and enhancing adaptability.

Among training-based methods, previous studies can be grouped into Pre-LLM, In-LLM, and joint approaches, according to where the reduction is applied. (1) Pre-LLM approaches compress tokens before the LLM via compact projectors (Cha et al., 2024; Li et al., 2024b) or encoder-side modules (Hu et al., 2024; Song et al., 2025; Zhang et al., 2025). Such approaches remain disconnected from the LLM's internal reasoning, preventing compression from adapting to cross-modal interactions. (2) In-LLM approaches integrate compression into the LLM, enabling strategies for token selection, aggregation, or reduction. Some methods perform representation compression by replacing vision tokens with latent tokens (Ye et al., 2024b) or by pooling operations (Chen et al., 2024a), while others adopt selection-based pruning, either through heuristic schedules (Xing et al., 2024; Shao et al., 2025; Liu et al., 2026) or adaptive strategies (Ye et al., 2024a). However, most pruning approaches rely on non-differentiable Top-$k$ operators, hindering end-to-end optimization. Dynamic-LLaVA (Huang et al., 2024) relaxes this with soft gating but still provides only approximate gradients, whereas our differentiable Top-$k$ yields a continuous relaxation with stable gradient flow. (3) Joint approaches combine the strengths of both Pre-LLM and In-LLM strategies, e.g., FocusLLaVA (Zhu et al., 2024), which applies vision-guided pre-LLM compression and text-guided pruning inside the LLM. While such hybrid designs demonstrate the potential of combining both perspectives, their two-stage pipeline increases architectural complexity and prevents unified end-to-end optimization. Our work instead focuses on the In-LLM setting, aiming to achieve effective compression with a fully differentiable and text-aware token selection strategy.

## C  LLM BACKBONES

Table 7: Detailed settings of LLM backbones.

| Model | Blocks | Heads | Hidden Dim | FFN Dim |
|---|---|---|---|---|
| MobileLLaMA 2.7B | 32 | 32 | 2560 | 6912 |
| Vicuna-7B-v1.5 | 32 | 32 | 4096 | 11008 |
| Vicuna-13B-v1.5 | 40 | 40 | 5120 | 13824 |

We use two decoder-only LLM backbones within the LLaVA-1.5 framework: MobileLLaMA 2.7B (Wu et al., 2024) and Vicuna-7B-v1.5 (Zheng et al., 2023). As shown in Table 7, both have 32 transformer blocks and 32 attention heads. MobileLLaMA 2.7B uses a hidden size of 2560 with an FFN dimension of 6912, while Vicuna-7B-v1.5 uses 4096 and 11008, respectively. Unless otherwise noted, all other architectural and training settings follow LLaVA defaults; our method changes only the vision token schedule and leaves the tokenizer, projector, and attention kernels untouched.

## D  BENCHMARKS

We conduct experiments on 11 mainstream benchmarks, including MME-Perception (Fu et al., 2023), MMBench, MMBench-CN (Liu et al., 2025), GQA (Hudson & Manning, 2019), VQAv2 (Goyal et al., 2017), ScienceQA-Iamge (Lu et al., 2022), VizWiz (Gurari et al., 2018), TextVQA (Singh et al., 2019), POPE (Li et al., 2023b), SEED-Image (Li et al., 2024a), and MM-Star (Chen et al., 2024c).

**MME-Perception** (Fu et al., 2023). A subset of tasks within the MME benchmark that focuses on evaluating a model's perception abilities. It relies on manually constructed instruction–answer pairs to ensure that the model must genuinely "understand" the image or comprehend the text to respond, rather than relying on memory or data leakage.

**MMBench** (Liu et al., 2025). Comprehensively measures a model's performance across different ability dimensions. It not only assesses whether the model can "understand" images or text but also evaluates its reasoning ability, knowledge integration, and more refined cognitive performance.

**GQA** (Hudson & Manning, 2019). Used to evaluate a model's understanding and reasoning abilities on real images. It emphasizes scene understanding and logical reasoning, not just the recognition of individual objects.

**VQAv2** (Goyal et al., 2017). Evaluates a model's visual perception ability through open-ended questions. Its core objective is to test whether the model can understand the content of an image and provide reasonable answers based on the questions.

**ScienceQA-Iamge** (Lu et al., 2022). Aims to evaluate a model's multimodal understanding, complex reasoning, and explainability abilities, covering multiple domains including natural sciences, language sciences, and social sciences.

**VizWiz** (Gurari et al., 2018). Used to evaluate a model's visual understanding under real-world, non-ideal image conditions. Its goal is to test whether the model can provide accurate answers in low-quality images and real-world question scenarios.

**TextVQA** (Singh et al., 2019). Focuses on evaluating a model's ability to understand textual information in images. It requires the model to recognize, read, and reason about the text in the image, and then generate correct answers by integrating visual information.

**POPE** (Li et al., 2023b). Used to evaluate the degree of object hallucination in models. Its core objective is to quantify the extent to which a model produces hallucinations, helping researchers understand the model's reliability in visual perception and generation.

**SEED-Image** (Li et al., 2024a). Evaluates a multimodal large model's ability to understand and generate image content. Its goal is to test the model's comprehensive multimodal abilities in visual perception, spatial reasoning, and image–text interaction tasks.

**MMStar** (Chen et al., 2024c). Aims to address insufficient visual dependency and data leakage issues in current multimodal evaluations. It defines 6 core visual–language (VL) abilities and constructs 18 detailed evaluation dimensions based on them, covering multiple aspects from coarse perception to fine-grained reasoning.

*Protocol.* Unless otherwise noted, we follow the official LLaVA evaluation protocol for all benchmarks above; MMStar is evaluated via `LMMS-Eval`.

## E  INTRODUCTION TO BASELINES

We conduct comparisons under the LLaVA-v1.5 (Liu et al., 2023a) framework to ensure consistency and fairness across different approaches. Specifically, we evaluate our method alongside several representative vision compression techniques, including FastV (Chen et al., 2024b), PDrop (Xing et al., 2024), VoCo-LLaMA (Ye et al., 2024b) and TwigVLM (Shao et al., 2025).

**FastV** (Chen et al., 2024b). A general plug-and-play method that prunes unnecessary visual tokens in the early filtering layer according to attention score ranking, thereby significantly reducing inference cost without sacrificing performance.

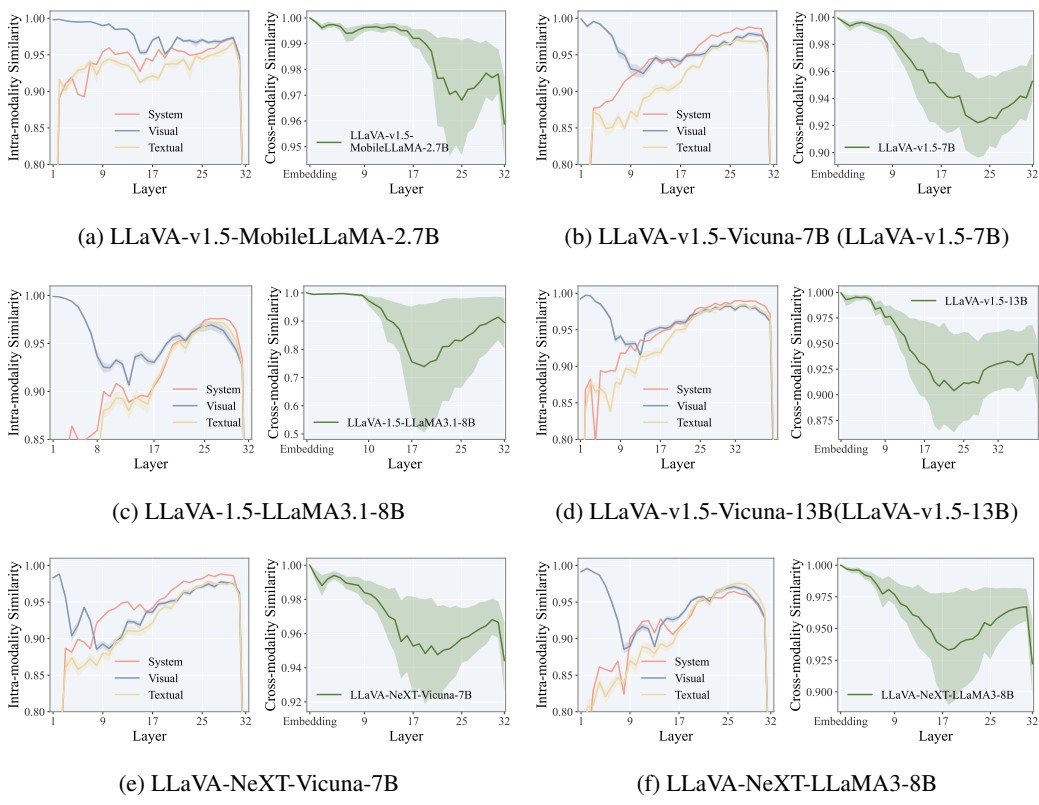

Figure 9: Layer-wise representational dynamics, where each subfigure consists of a left panel showing intra-modal refinement and a right panel highlighting cross-modal interaction intensity

**PDrop** (Xing et al., 2024). An approach dividing the LVLM into several stages, discarding part of the image tokens at the end of each stage based on lightweight similarity computation with a predefined ratio, with negligible time overhead.

**VoCo-LLaMA** (Chen et al., 2024b). The first method to compress visual information using LLMs, distilling the LLM's understanding of visual tokens into compact representations, compressing hundreds of visual tokens into a single VoCo token while minimizing information loss.

**TwigVLM** (Shao et al., 2025). A method that trains a lightweight twig block on the early layers of the base VLM, and through a twig-guided token pruning (TTP) strategy and a self-speculative decoding (SSD) strategy, achieves better accuracy and faster generation.

# F    EXTENDED ANALYSIS

## F.1    LAYER-WISE REPRESENTATIONAL DYNAMICS ANAYLSIS

To demonstrate that the phenomena observed in this paper are universal, we conducted layer-wise representational dynamics analysis on various LLM backbones and model sizes within the LLaVA-v1.5 framework, including MobileLLaMA-2.7B, Vicuna-7B, LLaMA3.1-8B and Vicuna-13B. As shown in Figs. 9a- 9d, all these LLMs exhibit similar trends and behaviors: (1) the intra-modality similarity in shallow layers starts at a relatively high level, then decreases and remains low for a while, before gradually increasing again and stabilizing at a higher level; and (2) the cross-modality similarity is also relatively high in the shallow layers. Besides, we also performed the same analysis under the LLaVA-NeXT framework using Vicuna-7B and LLaMA3-8B as backbones, as shown in Figs. 9e and  9f, and observed highly consistent patterns in both intra-modality and cross-modality similarity across layers. This further supports the universality of the identified phenomena across different LLM architectures and multimodal training pipelines.

Table 8: Complete per-benchmark results corresponding to Fig. 7. $I_i$ denotes visual injection at layer $i$, $E_i$ denotes early visual exit at layer $i$

| # | Model | $MME^P$ | MMB | $MMB^{CN}$ | GQA | $VQA^{v2}$ | $SQA^I$ | VizWiz | TextVQA | POPE | $SEED^I$ | MMStar | Avg(%) |
|---|---|---|---|---|---|---|---|---|---|---|---|---|---|
| 1 | Baseline | 1506.5 | 64.7 | 58.1 | 61.9 | 78.5 | 69.5 | 50.1 | 58.2 | 86.8 | 66.2 | 33.7 | 100.0 |
| | | | | | *Late Injection* | | | | | | | | |
| 2 | $I_5$ | 1441.9 | 66.0 | 59.9 | 62.4 | 78.5 | 69.5 | 51.9 | 55.9 | 86.5 | 65.6 | 34.0 | 100.1 |
| 3 | $I_6$ | 1442.5 | 65.8 | 58.3 | 62.4 | 78.5 | 69.3 | 51.5 | 56.5 | 86.8 | 66.1 | 33.3 | 99.7 |
| 4 | $I_7$ | 1413.8 | 66.2 | 58.8 | 62.2 | 78.3 | 69.8 | 48.6 | 56.6 | 86.4 | 65.3 | 33.1 | 99.0 |
| 5 | $I_8$ | 1424.1 | 65.1 | 58.2 | 62.7 | 78.3 | 69.1 | 50.7 | 57.3 | 87.1 | 65.9 | 31.9 | 99.1 |
| 6 | $I_9$ | 1444.4 | 65.4 | 57.9 | 61.5 | 77.9 | 68.9 | 53.0 | 56.1 | 86.5 | 65.3 | 32.7 | 99.3 |
| 7 | $I_{10}$ | 1402.3 | 63.5 | 54.4 | 62.0 | 77.8 | 68.9 | 50.6 | 56.8 | 87.0 | 63.7 | 32.7 | 97.7 |
| 8 | $I_{11}$ | 1392.8 | 63.1 | 56.4 | 61.9 | 77.7 | 68.6 | 50.9 | 57.6 | 86.8 | 62.8 | 33.3 | 97.7 |
| | | | | | *Fixed-Injection Span* | | | | | | | | |
| 9 | $I_9$ & $E_{22}$ | 1456.2 | 63.8 | 56.4 | 61.9 | 78.1 | 68.1 | 50.9 | 57.6 | 86.9 | 64.3 | 31.8 | 98.4 |
| 10 | $I_9$ & $E_{23}$ | 1438.2 | 63.9 | 56.6 | 61.9 | 78.1 | 66.7 | 51.9 | 58.0 | 86.7 | 65.0 | 32.9 | 98.7 |
| 11 | $I_9$ & $E_{24}$ | 1461.5 | 63.9 | 57.4 | 61.7 | 78.1 | 68.7 | 51.2 | 57.8 | 86.7 | 64.1 | 31.9 | 98.7 |
| 12 | $I_9$ & $E_{25}$ | 1436.6 | 65.8 | 56.4 | 62.5 | 77.9 | 67.1 | 51.5 | 57.3 | 87.1 | 65.3 | 33.5 | 99.2 |
| 13 | $I_9$ & $E_{26}$ | 1460.8 | 65.4 | 57.9 | 62.2 | 78.1 | 68.8 | 50.9 | 57.4 | 87.1 | 65.0 | 35.2 | 100.0 |
| 14 | $I_9$ & $E_{27}$ | 1435.9 | 65.2 | 58.2 | 62.4 | 78.0 | 68.5 | 48.3 | 56.7 | 86.9 | 64.9 | 33.1 | 98.6 |
| 15 | $I_9$ & $E_{28}$ | 1467.2 | 65.2 | 57.8 | 62.4 | 78.0 | 68.3 | 50.7 | 56.2 | 87.2 | 65.0 | 33.0 | 99.2 |
| | | | | | *Equal-Depth Window* | | | | | | | | |
| 16 | $I_8$ & $E_{24}$ | 1441.5 | 64.7 | 56.2 | 61.7 | 78.1 | 68.0 | 50.1 | 57.7 | 87.2 | 65.0 | 34.7 | 99.1 |
| 17 | $I_9$ & $E_{25}$ | 1436.6 | 65.8 | 56.4 | 62.5 | 77.9 | 67.1 | 51.5 | 57.3 | 87.1 | 65.3 | 33.5 | 99.2 |
| 18 | $I_{10}$ & $E_{26}$ | 1383.4 | 62.4 | 53.3 | 61.6 | 77.8 | 68.1 | 51.3 | 56.8 | 86.7 | 63.1 | 30.6 | 96.6 |

# G EXTENDED EXPERIMENTAL RESULTS

## G.1 LATE INJECTION AND EARLY EXIT

Our design of late injection and early exit is guided by two key diagnostics. First, layer 9 coincides with a local minimum in the visual layer-wise similarity curve (Fig. 2), suggesting a natural entry point for visual tokens. Second, accuracy plateaus around layer 25 under the deep-to-shallow masking experiment (Fig. 4), indicating a reasonable cutoff for discarding vision tokens. We validate these choices through three sets of sweeps (Fig. 7):

(1) *Late injection sweep.* Varying the injection layer while fixing the exit depth reveals a clear peak at layer 9. Injecting earlier increases computation with negligible gains, whereas injecting later leads to accuracy degradation.

(2) *Fixed-entry span sweep.* With injection fixed at layer 9, varying the exit depth yields an optimum around layers 25–26. Exiting later adds cost, while exiting earlier reduces accuracy.

(3) *Equal-depth window sweep.* Sliding a constant-length window confirms 8–24 and 9–25 as near-optimal spans, while 10–26 underperforms.

Notably, in the deep-to-shallow diagnostic, accuracy at layer 26 matches the baseline and at layer 25 is only marginally lower. We therefore select 25 as the exit depth, expecting training to recover the small gap. Taken together, these ablations validate the 9–25 window as a strong design choice for balancing efficiency and accuracy.

## G.2 DIFFERENTIABLE TOP-$k$

Here we present more detailed results on the advantages brought by our differentiable Top-$k$ operator. In the main text, we compared hard and differentiable Top-$k$ under a progressive pruning schedule (Table 3), showing that replacing hard Top-$k$ with differentiable Top-$k$ improves the av-

erage score from 97.7% to 99.7% with two-stage training (PT+FT) and from 97.5% to 98.1% with one-stage training (FT only).

Appendix Table 9 further demonstrates that the gain of differentiable Top-$k$ is most pronounced under high compression ratios. For example, when the number of visual tokens is reduced from the original 576 to as few as 72, hard Top-$k$ suffers clear degradation, whereas our differentiable Top-$k$ consistently preserves accuracy across benchmarks. The improvement is especially evident in vision-heavy tasks such as MMBench, SQA-I, and VizWiz, where more faithful token retention plays a critical role.

These results confirm that differentiable Top-$k$ provides a smoother selection mechanism that adapts to training signals, making it particularly effective in aggressive pruning regimes. We therefore adopt PT+FT with differentiable Top-$k$ as the default configuration in all main experiments.

Table 9: Performance comparison of LLaVA variants with Hard vs. Differentiable top-$k$ Operators. PT and FT denote pretrain and finetune, respectively.

| Model | Train | Topk | MME$^P$ | MMB | MMB$^{CN}$ | GQA | VQA$^{v2}$ | SQA$^I$ | VizWiz | TextVQA | POPE | SEED$^I$ | MMStar | Avg(%) |
|---|---|---|---|---|---|---|---|---|---|---|---|---|---|---|
| LLaVA-1.5-7B | - | - | 1506.5 | 64.7 | 58.1 | 61.9 | 78.5 | 69.5 | 50.1 | 58.2 | 86.8 | 66.2 | 33.7 | 100.0 |
| | | | | | | $576 \rightarrow 64 \rightarrow 8 \rightarrow 1$ | | | | | | | | |
| LLaVA-1.5-7B + TopK | PT+FT | Hard | 1436.9 | 64.2 | 57.0 | 59.7 | 76.4 | 70.4 | 50.1 | 55.7 | 86.5 | 63.1 | 33.6 | 98.0 |
| | | Diff. | 1484.7 | 65.5 | 56.3 | 60.2 | 76.3 | 71.5 | 52.7 | 56.2 | 86.2 | 63.3 | 34.3 | **99.3** |
| | FT | Hard | 1482.7 | 65.0 | 54.9 | 60.3 | 76.5 | 69.9 | 46.8 | 55.9 | 86.0 | 63.5 | 33.4 | 97.5 |
| | | Diff. | 1471.7 | 65.2 | 56.6 | 59.9 | 76.5 | 70.7 | 47.1 | 56.2 | 85.9 | 63.2 | 34.8 | **98.2** |

### G.3 TOKEN WEIGHTING STRATEGIES.

Table 10 reports the detailed results of different strategies for scoring visual tokens during training. We evaluate both *last-token* based methods, which compute importance by repeatedly attending from the last text token across multiple rounds, and *all-token* based methods, which aggregate attention from all text tokens to vision tokens. For each family, we also test variants that incorporate L2-norm weighting.

The results show that while all-token strategies slightly improve performance on some individual benchmarks, their overall average is not better than the multi-round last-token baseline. For example, the best all-token variant achieves 99.6% average, compared to 99.9% for the last-token (n-R) variant. Given the additional computational cost of eager attention required by all-token approaches, we conclude that the multi-round last-token scheme provides the best trade-off between efficiency and performance.

### G.4 POSITION ENCODING

Table 11 provides the detailed benchmark results of the three positional encoding (PE) schemes compared under the shallow–middle–deep compression setting. As discussed in the main text, the underlying challenge is conceptually similar to the "position-ID mismatch" in streaming LLMs (Tong et al., 2025b), but arises here from dynamic changes in the set of surviving vision tokens across layers due to late injection, progressive dropping, and early exit.

We evaluate three PE strategies: 1) *Persistent PE:* fixed RoPE indices assigned at input and never updated across layers. 2) *Compacted PE (PDrop-style):* indices are reset after pruning to compact surviving tokens and fill gaps. 3) *Group PE:* disjoint RoPE index ranges are allocated for text and vision tokens, avoiding in-place updates during token injection or removal.

As shown in Table 11, *Persistent PE achieves the highest average performance (97.8%)*, supporting the hypothesis that stable positional assignments mitigate cross-layer mismatch. *Group PE performs slightly worse (97.1%)*, suggesting that disjoint indexing is viable but not superior. By contrast, *Compacted PE yields the lowest accuracy (96.9%)*, confirming that index resets exacerbate position inconsistency. Given both its accuracy and zero additional overhead, we adopt Persistent PE as the default in all main experiments.

Table 10: Different strategies for scoring visual tokens. Last-token variants are computed using repeated attention from the last text token, while all-token variants aggregate attention from all text tokens, with or without L2-norm weighting.

| Model | MME$^P$ | MMB | MMB$^{CN}$ | GQA | VQA$^{v2}$ | SQA$^I$ | VizWiz | TextVQA | POPE | SEED$^I$ | MMStar | Avg(%) |
|---|---|---|---|---|---|---|---|---|---|---|---|---|
| LLaVA-1.5-7B | 1506.5 | 64.7 | 58.1 | 61.9 | 78.5 | 69.5 | 50.1 | 58.2 | 86.8 | 66.2 | 33.7 | 100.0 |
| Last token (1-R) | 1424.7 | 65.3 | 56.9 | 59.6 | 75.6 | 71.0 | 49.0 | 55.5 | 86.2 | 63.0 | 33.2 | 97.7 |
| Last token (n-R) | 1484.7 | 65.5 | 56.3 | 60.2 | 76.3 | 71.5 | 52.7 | 56.2 | 86.2 | 63.3 | 34.3 | 99.3 |
| Last token (n-R, L2) | 1447.0 | 65.2 | 56.8 | 59.7 | 76.3 | 70.6 | 48.8 | 55.9 | 86.5 | 63.5 | 34.0 | 98.2 |
| All token | 1414.8 | 65.0 | 59.2 | 59.0 | 74.8 | 70.3 | 51.4 | 56.6 | 86.4 | 63.4 | 34.3 | 98.6 |
| All token (L2) | 1424.0 | 65.5 | 58.7 | 59.9 | 75.2 | 68.9 | 53.2 | 56.6 | 87.0 | 64.7 | 35.5 | 99.6 |

Table 11: Effect of position encoding (PE) schemes under shallow–middle–deep compression. Persistent PE with fixed RoPE indices performs best overall, while resetting indices (Compacted PE) leads to accuracy degradation.

| Model | MME$^P$ | MMB | MMB$^{CN}$ | GQA | VQA$^{v2}$ | SQA$^I$ | VizWiz | TextVQA | POPE | SEED$^I$ | MMStar | Avg(%) |
|---|---|---|---|---|---|---|---|---|---|---|---|---|
| LLaVA-1.5-7B | 1506.5 | 64.7 | 58.1 | 61.9 | 78.5 | 69.5 | 50.1 | 58.2 | 86.8 | 66.2 | 33.7 | 100.0 |
| **Persistent PE** | 1414.4 | 63.7 | 56.7 | 61.3 | 76.6 | 67.0 | 52.1 | 55.6 | 86.9 | 65.2 | 32.0 | **97.8** |
| Compacted PE | 1452.3 | 64.6 | 56.1 | 61.1 | 76.8 | 67.9 | 48.9 | 55.1 | 86.5 | 64.6 | 30.3 | 96.9 |
| Group PE | 1442.2 | 63.9 | 55.4 | 60.4 | 76.2 | 67.6 | 51.2 | 55.5 | 86.9 | 63.6 | 31.1 | 97.1 |

## G.5 Filtering Layer Selection

Table 12 reports the detailed per-benchmark results for the selection of filtering layers. We first compute the ILVAS curve over the middle layers on a model configured with late injection and early exit. As shown in Figure 10, the ILVAS profiles are consistent across Top-$K \in \{5, 10, 20, 50, 100, 200\}$ and window sizes $n \in \{4, 8\}$, with local maxima occurring at layers $10, 14, 16, 18$. We therefore select $\{10, 14, 16, 18\}$ as the filtering layer set $\mathcal{F}$.

To validate this choice, we fix the concave pyramid token–decay schedule and sweep different layer configurations. Compared with a control schedule $\{12, 15, 18, 21\}$, the ILVAS-based selection achieves consistently higher average accuracy. Fixing $\{10, 16, 18\}$ and sweeping the remaining slot yields a clear peak at 14, whereas 12 or 13 lead to noticeable degradation. Similarly, joint sweeps of the middle pair confirm $\{14, 18\}$ as the strongest combination, while nearby alternatives such as $\{13, 18\}$, $\{13, 19\}$, and $\{14, 19\}$ underperform. These ablations confirm $\{10, 14, 16, 18\}$ as our final filtering-layer configuration for all main experiments, balancing efficiency and accuracy across tasks.

Table 12: Per-benchmark results for different filtering layer configurations under the concave pyramid dropping policy. The ILVAS-based set $\{10, 14, 16, 18\}$ achieves the best trade-off.

| # | Model | MME$^P$ | MMB | MMB$^{CN}$ | GQA | VQA$^{v2}$ | SQA$^I$ | VizWiz | TextVQA | POPE | SEED$^I$ | MMStar | Avg(%) |
|---|---|---|---|---|---|---|---|---|---|---|---|---|---|
| 1 | Baseline | 1506.5 | 64.7 | 58.1 | 61.9 | 78.5 | 69.5 | 50.1 | 58.2 | 86.8 | 66.2 | 33.7 | 100.0 |
| 2 | {12,15,18,21} | 1431.0 | 64.6 | 59.5 | 61.4 | 77.4 | 67.4 | 46.5 | 56.2 | 86.7 | 65.3 | 32.0 | 97.7 |
| 3 | {10,12,16,18} | 1452.9 | 61.3 | 55.2 | 60.7 | 76.8 | 67.6 | 49.3 | 54.1 | 86.4 | 64.5 | 31.7 | 96.5 |
| 4 | {10,13,16,18} | 1459.3 | 64.8 | 56.8 | 60.0 | 76.3 | 68.1 | 49.3 | 55.3 | 86.6 | 64.5 | 29.9 | 96.9 |
| 5 | {10,14,16,18} | 1469.5 | 65.0 | 56.2 | 60.9 | 76.7 | 69.0 | 50.8 | 55.1 | 86.1 | 64.7 | 33.1 | **98.3** |
| 6 | {10,15,16,18} | 1468.9 | 64.9 | 57.0 | 61.6 | 77.2 | 68.6 | 50.0 | 56.2 | 86.8 | 64.5 | 30.6 | 97.9 |
| 7 | {10,13,16,18} | 1459.3 | 64.8 | 56.8 | 60.0 | 76.3 | 68.1 | 49.3 | 55.3 | 86.6 | 64.5 | 29.9 | 96.9 |
| 8 | {10,13,16,19} | 1460.9 | 63.6 | 56.6 | 60.8 | 76.6 | 67.9 | 50.1 | 54.8 | 86.6 | 64.6 | 31.8 | 97.5 |
| 9 | {10,14,16,18} | 1469.5 | 65.0 | 56.2 | 60.9 | 76.7 | 69.0 | 50.8 | 55.1 | 86.1 | 64.7 | 33.1 | **98.3** |
| 10 | {10,14,16,19} | 1472.6 | 64.0 | 57.2 | 60.5 | 76.8 | 68.5 | 47.5 | 55.1 | 86.2 | 64.6 | 31.5 | 97.2 |

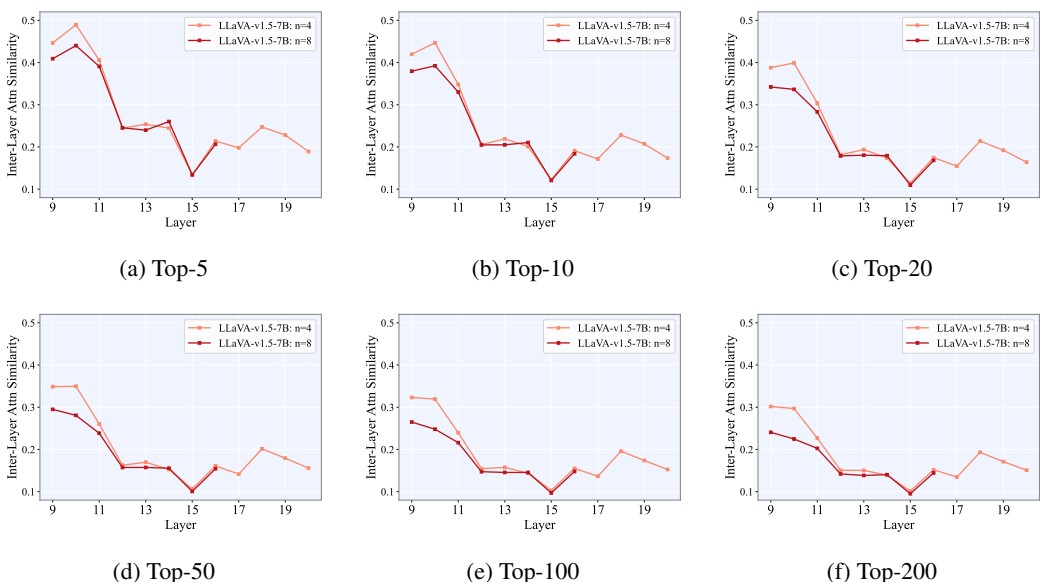

Figure 10: ILVAS curve over the middle layers on a model configured with the late injection and early exit. (a)–(f) sweep top-$k \in \{5, 10, 20, 50, 100, 200\}$, and each curve compares observation windows $n = 4$ and $n = 8$. Consistent valleys across $K$ indicate layers with strong filtering ability, i.e., candidates for the pruning set $\mathcal{F}$.

These results support our final choice of $\{10, 14, 16, 18\}$ as the filtering layers for all main experiments, balancing efficiency and accuracy across tasks.

