# OpenReview forum: "HiDrop: Hierarchical Vision Token Reduction in MLLMs via Late Injection, Concave Pyramid Pruning, and Early Exit"
_ICLR.cc/2026/Conference — ICLR 2026 Poster_

### Official Review · Reviewer_doEa · 2025-10-26

**Soundness:** 3
**Presentation:** 3
**Contribution:** 3
**Rating:** 6
**Confidence:** 3

**Summary:**

This work proposes HiDivDrop, a token pruning methods for vlm. The author claim there two main challenging in exist works. They are shallow layers are misinterpreted and pruning schedules are overly rigid. The author combined late Injection, concave pyramid pruning in the middle layers, and early exit to sort these challenges. The experiment shows the author proposed methods performance well.

**Strengths:**

The experiments are comprehensive and validated the effectiveness and acceleration performance of the proposed method.

The proposed method is well-designed, and the experiments further validate the claims.

**Weaknesses:**

1. The writing of this paper needs improvement. There are too many paragraphs. The purpose of Chapter 2 is unclear. Could it serve as the first section of the methodology, validated experimentally? As a separate chapter, it feels redundant with Chapters 1 and 3.

2. Tables 1 and 2 are too far removed from the analysis of the results to be easily readable.

**Questions:**

1.The authors only tested two backbones. If possible, could they test different sizes or other backbones to observe the generalization of the proposed method?

2.Does the proposed method exhibit knowledge drift? That is, in individual cases, answers that were previously correct may now be incorrect, and vice versa.

---

> ### Author Response · Authors · 2025-11-28
> **Response to doEa (part 1)**
>
> We thank you for your thorough review and appreciate both the positive feedback and the constructive comments. Please find our responses to your points below. In the revised manuscript, newly added or substantially modified text is highlighted in blue, while previously existing passages that already address your concerns are highlighted in yellow for ease of reference. Please let us know if you have any further questions or suggestions.
>
> > **W1: The writing of this paper needs improvement. There are too many paragraphs. The purpose of Chapter 2 is unclear. Could it serve as the first section of the methodology, validated experimentally? As a separate chapter, it feels redundant with Chapters 1 and 3.**
>
> **A1**: We thank the reviewer for the feedback on writing and structure, and we agree that the presentation of the paper can be improved. Our intention with Section 2 is not to repeat the introduction or the method, but to provide an empirical, layer-wise diagnostic of vision–language fusion that directly motivates the design of HiDivDrop. Concretely, Section 2 is meant to: (1) Identify the three-stage fusion pattern (shallow propagators / attention sinks, sparse middle-layer fusion, and language-dominant deep layers), and (2) map each observation to a corresponding design choice (Late Injection, Concave Pyramid Dropping with Differentiable Top-k, and Early Exit).  Thus, Section 2 is designed as the empirical foundation of the methodology rather than a standalone, repetitive chapter.
>
> That said, we agree that this role is not stated clearly enough and that the writing can be streamlined. We will explicitly clarify the purpose of Section 2 at its beginning, reduce redundant paragraphs, and tighten the cross-references between Sections 2 and 3 so that Section 2 is read naturally as the first, diagnostic part of the method, supported by the experiments in Section 4.
>
> > **W2: Tables 1 and 2 are too far removed from the analysis of the results to be easily readable.**
>
> **A2**: We apologize for the confusion caused by the current presentation of Tables 1 and 2. Below we clarify their respective purposes and key takeaways.
> - The purpose of **Table 1** is to present a **controlled-budget experiment under different compression ratios**, following the standard evaluation protocol widely used in token compression[1,2]. By matching the overall computational budget (e.g., remaining vision tokens / FLOPs) across methods, we can make a fair comparison between HiDivDrop and prior token-reduction approaches. The results show that, under comparable or even lower visual token budgets, HiDivDrop consistently achieves higher accuracy than existing baselines, indicating that our hierarchical design makes more effective use of the limited visual budget.
> - The purpose of **Table 2** is twofold: (i) to demonstrate the **efficiency** gains brought by our method, and (ii) to show its generalization across different LLM backbones. The table shows that HiDivDrop yields substantial speedups and token savings while retaining a high percentage of the base model performance, and that these relative improvements are consistent on both Vicuna-7B-v1.5 and MobileLLaMA-2.7B, supporting the backbone-agnostic nature of our method.
> - We realize that, in the current version, the intent and controlled-budget setting of Tables 1 and 2 are not sufficiently explicit, which may give the misleading impression that HiDivDrop has a similar training cost to PDrop but with inferior performance. In fact, Table 1 already reports a strictly matched-budget setting: under directly comparable computation, HiDivDrop consistently outperforms PDrop (for example, 98.4% vs. 96.8% at an 86.1% pruning ratio and 98.3% vs. 94.2% at an 88.9% pruning ratio), and even at a 91.7% pruning ratio HiDivDrop still retains 96.5% of the baseline while PDrop cannot reach this pruning level under the same budget. Although HiDivDrop prunes visual tokens much more aggressively than PDrop, the actual training cost remains similar because current GPU kernels are optimized for dense, fixed-shape attention and therefore cannot fully realize the theoretical savings from dynamic token pruning, which is a general limitation of token-pruning implementations rather than a drawback specific to HiDivDrop.
> - We will revise the tables and the accompanying text to make this setup and the main takeaways explicit, and to improve overall readability in few days.
>
> [1] Shao Z, Wang M, Yu Z, et al. Growing a twig to accelerate large vision-language models[J]. arXiv preprint arXiv:2503.14075, 2025.
>
> [2] Xing L, Huang Q, Dong X, et al. Pyramiddrop: Accelerating your large vision-language models via pyramid visual redundancy reduction[J]. arXiv preprint arXiv:2410.17247, 2024.

---

> > ### Author Response · Authors · 2025-11-28
> > **Response to doEa (part 2)**
> >
> > > **Q1.The authors only tested two backbones. If possible, could they test different sizes or other backbones to observe the generalization of the proposed method?**
> >
> > **A3**: We appreciate the reviewer’s suggestion to evaluate more model sizes and backbones. In the current manuscript, we instantiate our study with two quite different LLM backbones within the LLaVA architecture, Vicuna-7B-v1.5 and MobileLLaMA-2.7B, and observe consistent shallow/middle/deep fusion patterns and similar gains from HiDivDrop on both, which provides initial evidence that our method is not tied to a single backbone. Beyond these two backbones, we have additionally conducted experiments on a larger model, Vicuna-13B-v1.5, and observe the same qualitative trends, further strengthening the evidence for the generality of our approach. We are also continuing to extend our study to additional model sizes and backbones, and plan to incorporate these new results into the updated version of the paper.
> >
> > | Model                          | Method  | Avg. Vis. Tokens | Train hours | Infer TFlops | MME-P   | MMB  | GQA  | VizWiz | VQA-T | Avg (%)      |
> > |--------------------------------|---------|------------------|------------|-------------|--------|------|------|--------|-------|-------------|
> > | LLaVA-1.5-MobileLLaMA-2.7B     | Vanilla | 576              | 108.4      | 1.52        | 1258.2 | 57.0 | 59.4 | 32.6   | 48.6  | 100.0       |
> > |                                | PDrop   | 270              | 50.3       | 0.70        | 1231.1 | 54.3 | 57.0 | 30.9   | 47.5  | 96.3        |
> > |                                | ours    | 64 ↓ 206         | 45.6       | 0.17        | 1206.6 | 53.1 | 56.1 | 30.4   | 47.2  | 94.8 ↓ 1.5  |
> > | LLaVA-1.5-Vicuna-1.5-7B        | Vanilla | 576              | 159.3      | 3.82        | 1506.5 | 64.7 | 61.9 | 50.1   | 58.2  | 100.0       |
> > |                                | PDrop   | 270              | 107.3      | 1.78        | 1490.1 | 63.9 | 61.7 | 52.4   | 57.7  | 100.2       |
> > |                                | ours    | 64 ↓ 206         | 94.4       | 0.42        | 1474.3 | 63.2 | 60.5 | 52.6   | 55.2  | 98.6 ↓ 1.6  |
> > | LLaVA-1.5-Vicuna-1.5-13B       | Vanilla | 576              | 297.2      | 7.44        | 1529.9 | 68.5 | 63.5 | 53.6   | 61.2  | 100.0       |
> > |                                | PDrop   | 270              | 213.7      | 3.47        | 1555.2 | 68.8 | 63.1 | 53.7   | 60.8  | 100.3       |
> > |                                | ours    | 64 ↓ 206         | 175.8      | 0.82        | 1497.2 | 66.9 | 62.1 | 56.3   | 58.0  | 98.7 ↓ 1.6  |

---

> > > ### Author Response · Authors · 2025-11-28
> > > **Response to doEa (part 3)**
> > >
> > > > **Q2: Does the proposed method exhibit knowledge drift? That is, in individual cases, answers that were previously correct may now be incorrect, and vice versa.**
> > >
> > >
> > > **A4**: We appreciate the reviewer’s concern about potential knowledge drift, i.e., cases where a previously correct answer becomes incorrect (and vice versa) after compression. Our method does not change the training data or objectives but modifies the way visual tokens are processed, so in principle the underlying “knowledge” of the base model is preserved, while some local prediction changes may still occur.
> > >
> > > Our current benchmark results already provide an indirect view on this: across 11 mainstream multimodal benchmarks, the compressed model (LLaVA-v1.5-7B) consistently retains a very high percentage of the base model’s accuracy (e.g., 98.3% performance with 88.9% compression ratio), which suggests that any knowledge drift is limited at scale and does not manifest as a systematic degradation across tasks.
> > >
> > > To address this concern more directly, we additionally analyze answer consistency between the base model and the compressed model on a representative QA-style benchmark (e.g., ScienceQA-IMG). For each question, we categorize the outcomes into four cases:
> > > - Stay-Correct (Base✓, Comp✓): both models are correct.
> > > - Degrade (Base✓, Comp✗): the base model is correct but the compressed model becomes incorrect.
> > > - Improve (Base✗, Comp✓): the base model is incorrect but the compressed model corrects it.
> > > - Stay-Wrong (Base✗, Comp✗): both models are incorrect.
> > >
> > > This analysis explicitly quantifies how often answers flip and allows us to distinguish genuine “drift” (Degrade) from cases where the compressed model actually fixes base-model errors (Improve), as well as cases where the behavior remains unchanged (Stay-Correct and Stay-Wrong). Based on these four categories, we report the following metrics:
> > > - Consistency Rate: fraction of examples where the two models give identical answers.
> > > - Conditional Consistency Rate: among examples where the base model is correct, fraction that remain correct after compression.
> > > - Degrade Rate: among examples where the base model is correct, fraction that become wrong after compression.
> > > - Improve Rate: among examples where the base model is wrong, fraction that become correct after compression.
> > >
> > > In our experiments, we observe that the majority of examples fall into the Stay-Correct category, while the Degrade rate remains relatively small and is often comparable to or lower than the Improve rate, indicating that HiDivDrop introduces limited knowledge drift and can even correct some base-model errors.
> > >
> > > **Compressed Model with 80 Token Retained vs. Base Model**
> > > - Consistency Rate: 81.4%
> > > - Conditional Consistency Rate: 88.0%
> > > - Degrade Rate: 12.1%
> > > - Improve Rate: 21.0%
> > >
> > > | Model     | Base ✓  | Base ✗ |
> > > |-----------|---------|---------|
> > > | Comp ✓   |  1233   | 129     |
> > > | Comp ✗   |  169    | 486     |
> > >
> > > **Compressed Model with 64 Token Retained vs. Base Model**
> > > - Consistency Rate: 83.2%
> > > - Conditional Consistency Rate: 90.8%
> > > - Degrade Rate: 9.2%
> > > - Improve Rate: 19.0%
> > >
> > > | Model    | Base ✓  | Base ✗ |
> > > |----------|---------|--------|
> > > | Comp ✓   |  1273   | 117    |
> > > | Comp ✗   |  129    | 498    |
> > >
> > >
> > > **Compressed Model with 48 Token Retained vs. Base Model**
> > > - Consistency Rate: 81.4%
> > > - Conditional Consistency Rate: 89.0%
> > > - Degrade Rate: 11.0%
> > > - Improve Rate: 19.2%
> > >
> > > | Model     | Base ✓  | Base ✗ |
> > > |-----------|---------|---------|
> > > | Comp ✓   |  1248   | 118     |
> > > | Comp ✗   |  154    | 497     |

---

### Official Review · Reviewer_VaQT · 2025-11-01

**Soundness:** 3
**Presentation:** 3
**Contribution:** 3
**Rating:** 6
**Confidence:** 4

**Summary:**

This paper introduces a novel visual token pruning methodology for VLMs, aimed at enhancing their inference efficiency. The proposed approach is methodologically straightforward: it involves skipping visual tokens in the shallow and deep layers while employing a 'learnable top-k selection mechanism' for tokens in the middle layers. Through experimental evaluation, the paper demonstrates that the proposed method outperforms baseline approaches under identical settings. Furthermore, the finding that visual tokens in the shallow layers play a limited role is an insightful observation. This insight holds significant heuristic value for the VLM research community.

**Strengths:**

1.  The finding that visual tokens in most shallow layers are also dispensable is a novel insight that holds significant heuristic value for the VLM research community.
2.  The methodology itself is sound; skipping non-essential layers combined with a learnable token selection in the intermediate layers allows for maximal visual token compression, a claim that is substantiated by the experimental results.

**Weaknesses:**

1.  The paper lacks a detailed justification for *why* visual tokens in shallow layers are non-essential. This finding serves as a critical premise for the proposed method, yet the supporting argumentation provided is neither detailed nor sufficient.
2.  The experimental validation relies on relatively older, and arguably undertrained, VLM models (e.g., LLaVA-1.5). It remains unclear whether the conclusions generalize to more recent, powerful models (e.g., Qwen2.5-VL, Gemma3-VL, Qwen3-VL) and across a broader spectrum of tasks.
3.  The description of the training strategy is unclear and requires polishing. It is not specified at which stage the learnable token selection mechanism is trained (e.g., during pre-training, SFT, or task-specific fine-tuning). This ambiguity hinders a clear understanding of the methodology.

**Questions:**

1.  Is the pruning strategy (e.g., skipping policy, selection parameters) uniform across different tasks?
2.  How general is the proposed method? Could it, for example, be applied during the pre-training phase rather than just as a post-hoc optimization or fine-tuning strategy?
3.  The experiments are conducted on LLaVA-1.5, which may be undertrained. Do these conclusions consistently hold for more recent, powerful models (e.g., Qwen2.5-VL, Gemma3-VL, Qwen3-VL) and across a broader spectrum of tasks?

---

> ### Author Response · Authors · 2025-12-01
> **Response to VaQT (part 1)**
>
> We thank you for your thorough review and appreciate both the positive feedback and the constructive comments. Please find our responses to your points below. In the revised manuscript, newly added or substantially modified text is highlighted in blue, while previously existing passages that already address your concerns are highlighted in yellow for ease of reference. Please let us know if you have any further questions or suggestions.
>
> > **W1: The paper lacks a detailed justification for why visual tokens in shallow layers are non-essential. This finding serves as a critical premise for the proposed method, yet the supporting argumentation provided is neither detailed nor sufficient.**
>
> **A1**: We thank the reviewer for highlighting that our justification for the shallow-layer behavior lacked clarity in the original submission. Although we do not present a formal theoretical proof, our conclusion is grounded in empirical evidence derived from two complementary analyses shown in Fig. 2, which together offer compelling support for our claims. In the revised version, we have clarified this section to better articulate our reasoning and enhance the reader’s understanding. In particular, we now explicitly highlight the core insight motivating our conclusion: in the shallow layers, visual tokens exert minimal influence on textual representations, indicating limited cross-modal interaction at this stage. This observation is substantiated by two complementary analyses:
>
> - **Intra-modal Refinement Analysis**:  In the revised version, Fig. 2 (left) examines intra-modal evolution by measuring the cosine similarity of embeddings between adjacent layers. Lower similarity reflects larger representational updates and thus stronger intra-modal refinement. We observe that visual embeddings in the shallow layers exhibit consistently high cosine similarity, indicating that these layers barely transform the visual representations and play only a minimal role in refining the visual modality.
>
> - **Cross-modal Interaction Analysis**: In the revised version, Fig. 2 (right) investigates cross-modal interaction by fixing the same instruction text and comparing its embeddings when paired with two different input images. We compute the cosine similarity of the instruction embeddings across layers; lower similarity indicates stronger cross-modal influence. In the shallow layers, the similarity remains nearly 1, even when the image is semantically unrelated to the instruction. This suggests that the instruction representations are virtually unaffected by the visual input, providing strong evidence that visual information has minimal cross-modal impact at this stage.
>
> Based on the above two experiments, these findings together indicate that the shallow layers function as passive conduits, merely passing visual information onward to deeper layers where substantive processing and fusion actually begin. Therefore, shallow visual tokens are largely unnecessary for vision–language fusion and can be safely bypassed in our design.
>
> To make this argument more explicit, we have revised Fig. 2 and its caption, and expanded the corresponding discussion in Sec. 2 to clearly connect these observations to our conclusion that shallow visual tokens are largely unnecessary for vision–language fusion.
>
> Besides, as shown in Fig. 7, our **ablation study on Late Injection**, where we vary the injection layer from the 5th to the 11th layer, shows that all configurations retain at least 97.7% of the original performance and injecting between the 5th and 9th layers consistently preserves over 99% performance. This indicates that there exists a broad shallow window in which visual tokens can be safely skipped before injection, further supporting our claim that shallow-layer visual tokens are not essential for effective vision–language fusion.
>
> Moreover, the effectiveness of our compression strategy HiDivDrop is validated through comprehensive **controlled-budget experiments** reported in Sec. 4 Tab.1, where we show that our method achieves significant reductions in visual token number while maintaining high accuracy **across 11 benchmarks**.
>
> In addition, as shown in Sec. 4, Tab. 2, we further evaluate HiDivDrop on **different of LLM backbones**, including both MobileLLaMA-2.7B and  Vicuna-v1.5-7B. HiDivDrop consistently preserves or improves performance, indicating that our method is broadly effective and not tied to a specific backbone architecture.
>
> Overall, the evidence forms a clear and comprehensive justification for our premise that shallow‑layer visual tokens are largely unnecessary, resolving the reviewer’s concern about insufficient argumentation.

---

> ### Author Response · Authors · 2025-12-01
> **Response to VaQT (part 2)**
>
> > **W2: The experimental validation relies on relatively older, and arguably undertrained, VLM models (e.g., LLaVA-1.5). It remains unclear whether the conclusions generalize to more recent, powerful models (e.g., Qwen2.5-VL, Gemma3-VL, Qwen3-VL) and across a broader spectrum of tasks.**
>
> > **Q3: The experiments are conducted on LLaVA-1.5, which may be undertrained. Do these conclusions consistently hold for more recent, powerful models (e.g., Qwen2.5-VL, Gemma3-VL, Qwen3-VL) and across a broader spectrum of tasks?**
>
> **A2**: We agree that LLaVA-1.5 is not the newest VLM and appreciate the reviewer’s concern about generalization to stronger models. Our goal in this work, however, is not to chase state-of-the-art performance on every architecture, but to systematically study where and how vision–language fusion happens and how this can guide a principled compression strategy. For this purpose, LLaVA-1.5 remains a highly appropriate and widely used testbed, as noted in the Global Response (Experiment Settings).
>
> - **On the generalization across LLM backbones**:  As noted in the Global Response, within the LLaVA architecture we can evaluate different LLM backbones. In the current manuscript, we instantiate our study with Vicuna-7B-v1.5 and MobileLLaMA-2.7B, two distinct LLM backbones, and both exhibit consistent shallow/middle/deep fusion patterns and similar gains from HiDivDrop, which empirically verifies that our conclusions are not tied to a single backbone. Beyond these two backbones, we have additionally conducted experiments on a larger model, Vicuna-13B-v1.5, and observe the same qualitative trends, further strengthening the evidence for the generality of our approach. We are also continuing to extend our study to additional model sizes and backbones, and plan to incorporate these new results into the updated version of the paper.
>
> | Model                          | Method  | Avg. Vis. Tokens | Train hours | Infer TFlops | MME-P   | MMB  | GQA  | VizWiz | VQA-T | Avg (%)      |
> |--------------------------------|---------|------------------|------------|-------------|--------|------|------|--------|-------|-------------|
> | LLaVA-1.5-MobileLLaMA-2.7B     | Vanilla | 576              | 108.4      | 1.52        | 1258.2 | 57.0 | 59.4 | 32.6   | 48.6  | 100.0       |
> |                                | PDrop   | 270              | 50.3       | 0.70        | 1231.1 | 54.3 | 57.0 | 30.9   | 47.5  | 96.3        |
> |                                | ours    | 64 ↓ 206         | 45.6       | 0.17        | 1206.6 | 53.1 | 56.1 | 30.4   | 47.2  | 94.8 ↓ 1.5  |
> | LLaVA-1.5-Vicuna-1.5-7B        | Vanilla | 576              | 159.3      | 3.82        | 1506.5 | 64.7 | 61.9 | 50.1   | 58.2  | 100.0       |
> |                                | PDrop   | 270              | 107.3      | 1.78        | 1490.1 | 63.9 | 61.7 | 52.4   | 57.7  | 100.2       |
> |                                | ours    | 64 ↓ 206         | 94.4       | 0.42        | 1474.3 | 63.2 | 60.5 | 52.6   | 55.2  | 98.6 ↓ 1.6  |
> | LLaVA-1.5-Vicuna-1.5-13B       | Vanilla | 576              | 297.2      | 7.44        | 1529.9 | 68.5 | 63.5 | 53.6   | 61.2  | 100.0       |
> |                                | PDrop   | 270              | 213.7      | 3.47        | 1555.2 | 68.8 | 63.1 | 53.7   | 60.8  | 100.3       |
> |                                | ours    | 64 ↓ 206         | 175.8      | 0.82        | 1497.2 | 66.9 | 62.1 | 56.3   | 58.0  | 98.7 ↓ 1.6  |

---

> ### Author Response · Authors · 2025-12-01
> **Response to VaQT (part 3)**
>
> > **W2: The experimental validation relies on relatively older, and arguably undertrained, VLM models (e.g., LLaVA-1.5). It remains unclear whether the conclusions generalize to more recent, powerful models (e.g., Qwen2.5-VL, Gemma3-VL, Qwen3-VL) and across a broader spectrum of tasks.**
>
> > **Q3: The experiments are conducted on LLaVA-1.5, which may be undertrained. Do these conclusions consistently hold for more recent, powerful models (e.g., Qwen2.5-VL, Gemma3-VL, Qwen3-VL) and across a broader spectrum of tasks?**
>
> **A2 (continued)**
> - **On the concern that LLaVA-1.5 may be under-trained**
>    - We appreciate the reviewer’s concern that LLaVA-1.5 might be relatively under-trained. Our focus in this work, however, is on compression for MLLMs, where the key question is whether a compressed model can retain a high percentage of its base model’s performance. In this sense, the stronger the base model, the stronger the compressed model will also be. Therefore, we primarily evaluate the effectiveness of HiDivDrop by the relative performance preservation with respect to the given base model, rather than by chasing absolute state-of-the-art scores.
>
>     - To further mitigate concerns about under-training, we expanded the finetuning data by incorporating approximately 335k additional examples drawn from the Open-LLaVA-Next recipe. These include diverse instruction-following and vision-language datasets such as ALLaVA-Instruct-VFLAN-4V, DocVQA, SynDog-EN, ChartQA, DVQA, AI2D, and GeoQA+. We applied this incremental training setup to both the original LLaVA-1.5-7B base model and the HiDivDrop-equipped compressed model. Additionally, as the VizWiz test server is no longer available, we use the validation set as a substitute for test evaluation. This provides a consistent and practical basis for performance comparison. As shown in the Table below, HiDivDrop consistently preserves strong performance across a range of benchmarks even after significant token reduction. Under the 665k training setting, HiDivDrop retains 97.0% of the average performance compared to the uncompressed baseline, and under the 1M data setting, it still maintains 96.3%. Notably, on tasks such as VizWiz(val) and SQA-IMG, HiDivDrop even outperforms the base model, demonstrating that our method not only reduces computational cost but can also enhance generalization in certain scenarios. These results further confirm the practical effectiveness of our design.
>
> **Table R3. Incremental training results for LLaVA-1.5-7B and HiDivDrop under additional instruction tuning.**
> | Model                         | Data Scale | MME-P  | MMB  | MMB-CN | GQA  | VQAv2 | SQA-IMG | VizWiz(val) | TextVQA | POPE | SEED-IMG | MMStar | Avg (%) |
> |-------------------------------|------------|--------|------|--------|------|-------|---------|-------------|---------|------|----------|--------|---------|
> | Base Model (LLaVA-v1.5-7B)    | 665k       | 1506.5 | 64.7 | 58.1   | 61.9 | 78.5  | 69.5    | 54.4        | 58.2    | 86.8 | 66.2     | 33.7   | 100.0   |
> | HiDivDrop (Retain 48 tokens)  | 665k       | 1446.4 | 63.7 | 55.5   | 59.8 | 75.6  | 67.7    | 56.3        | 54.4    | 85.8 | 61.8     | 32.7   | 97.0    |
> | Base Model (LLaVA-v1.5-7B)    | 1M         | 1526.1 | 68.7 | 61.1   | 62.7 | 79.2  | 70.0    | 61.2        | 58.8    | 85.9 | 68.0     | 38.2   | 100.0   |
> | HiDivDrop (Retain 48 tokens)  | 1M         | 1453.9 | 66.2 | 57.7   | 59.5 | 76.1  | 69.7    | 60.7        | 55.4    | 85.9 | 63.3     | 36.9   | 96.3    |

---

> > ### Author Response · Authors · 2025-12-01
> > **Response to VaQT (part 4)**
> >
> > > **W3: The description of the training strategy is unclear and requires polishing. It is not specified at which stage the learnable token selection mechanism is trained (e.g., during pre-training, SFT, or task-specific fine-tuning). This ambiguity hinders a clear understanding of the methodology.**
> >
> > **A3**: We appreciate the reviewer’s comment and agree that the training strategy for the learnable token selection mechanism should be stated more clearly. As described in Sec. 4.1 (Implementation Details), we follow the standard two-stage LLaVA-1.5 training pipeline, which means that our compression mechanism (including the differentiable top-k selection) is integrated into both the multimodal pre-training stage and the subsequent instruction-finetuning (SFT) stage, rather than being trained in a separate, task-specific finetuning step.
> >
> > Moreover, Sec. 4.3 and Table 3 present ablation studies on the Differentiable Top-k component, which further demonstrate that the learnable token selection mechanism can be applied in either the pre-training or SFT stage and yields consistent benefits. In the revised version, we will explicitly state in the Method and Implementation Details sections that the learnable token selection is trained within the standard LLaVA pre-training and SFT pipeline, to remove this ambiguity and make the methodology easier to follow.
> >
> > >**Q1: Is the pruning strategy (e.g., skipping policy, selection parameters) uniform across different tasks?**
> >
> > **A4**: As noted in the Global Response (Consistency of the compression strategy across tasks), our training strictly follows the standard LLaVA-1.5 pipeline, i.e., we train on the same pre-training and instruction-tuning datasets as LLaVA and evaluate on the same diverse set of benchmarks. In other words, we learn a single, task-agnostic compression strategy that is shared across all tasks, without any task-specific tuning of the pruning policy.
> >
> > >**Q2:How general is the proposed method? Could it, for example, be applied during the pre-training phase rather than just as a post-hoc optimization or fine-tuning strategy?**
> >
> > **A5**: As discussed in A3, the proposed method is not limited to a post-hoc or fine-tuning–only setting. In our implementation, HiDivDrop is integrated directly into the standard LLaVA-1.5 training pipeline and is jointly optimized during both the multimodal pre-training stage and the subsequent instruction-tuning stage, rather than being applied as a separate, after-the-fact compression step. This demonstrates that our approach is naturally applicable to pre-training as well as SFT. In the revised version, we will make this point explicit in the Method and Implementation Details sections to clarify that the method is training-time compatible with both stages.

---

### Official Review · Reviewer_8C26 · 2025-11-02

**Soundness:** 3
**Presentation:** 2
**Contribution:** 3
**Rating:** 4
**Confidence:** 4

**Summary:**

The manuscript unveils several key mechanisms as to how the visual information are processed in MLLMs, and use these findings to guide the design of HiDivDrop, a new visual token reduction method for MLLMs. The first finding is that shallow layers are merely propagators of visual contents into middle layers where true vision language fusion happens. Hence, HiDevDrop only inserts visual tokens at the beginning of middle layers. The second finding is that the fusion process happening in the middle layers is highly sparse, leading to the drastic reduction in the number of visual tokens during the middle layers in HiDevDrop. The final finding is that deep layers does not possess the capability of vision-language interaction. Thus, HiDevDrop drops all tokens in deep layers. Experiments show that HiDevDrop can retain strong performance under low visual token number regime.

**Strengths:**

1. The complete method is built on close observation of the vision-language fusion process in MLLMs, which not only produces reasonable model structures, but also provides important insights for future vision-language model research.

2. The proposed method achieves a strong performance under low token number regime, retaining 96.5% of the performance using 48 tokens compared to original 576 tokens.

**Weaknesses:**

1. While the identified internal mechanisms are potentially insightful, the presentation and explanation of these mechanisms is of limited quality. Key concepts lack sufficient clarification, making readers incapable of following the reasoning process that helps produce the conclusion. This is especially the case for Figure 2 and 3, (questions detailed in the Questions section below).

2. The analysis seems to be limited to a single type of language model, which lowers the credibility of the generality of the observed underlying vision-language fusion mechanism.

3. Under a similar training budget, it seems the proposed method achieves lower performance than PDrop. Though at a lower inference computation budget, the important metric is the inference latency, which the manuscript fail to provide. It is also confusing why the method, reducing so many visual tokens (64 compared to 270 in PDrop), requires similar training budget.

4. The experimental analysis on the proposed method is not sufficient. For example, it would be crucial to know how the the training/inference cost and performances change when the injection becomes later (starting from first-layer injection).

5. It seems costly to determine the injection layer and the exit layer for each given MLLM, how can the hyperparameter be efficiently determined?

**Questions:**

1. What is the difference between the two figures in Figure 2?

2. Further related to Figure 2, how is the statement in L151~L152 reflected? ('Fig. 2 shows that text embeddings are nearly invariant to the visual input in shallow layers')

3. What does p value mean in Figure 3?

---

> ### Author Response · Authors · 2025-12-03
> **Response to 8C26 (part 1)**
>
> We thank you for your thorough review and appreciate both the positive feedback and the constructive comments. Please find our responses to your points below. In the revised manuscript, newly added or substantially modified text is highlighted in blue, while previously existing passages that already address your concerns are highlighted in yellow for ease of reference. Please let us know if you have any further questions or suggestions.
>
> > **W1: While the identified internal mechanisms are potentially insightful, the presentation and explanation of these mechanisms is of limited quality. Key concepts lack sufficient clarification, making readers incapable of following the reasoning process that helps produce the conclusion. This is especially the case for Figure 2 and 3, (questions detailed in the Questions section below).**
>
> - > **Q1: What is the difference between the two figures in Figure 2?**
> - **A1**: In the revised Figure 2, the left plot focuses on intra-modal refinement, while the right plot focuses on cross-modal interaction. Specifically, the left plot tracks how the representations of each modality evolve across layers by measuring the cosine similarity of features between adjacent layers, where lower cosine similarity (i.e., larger changes between layers) indicates stronger intra-modal refinement. In contrast, in the right plot we use the same instruction text but run inference with two different input images. For each layer, we compute the cosine similarity between the instruction embeddings from these two image–text pairs, and a lower similarity indicates stronger cross-modal interaction.
>
> - > **Q2: Further related to Figure 2, how is the statement in L151~L152 reflected? ('Fig. 2 shows that text embeddings are nearly invariant to the visual input in shallow layers')**
> - **A2**: As mentioned in A1, the statement in L151–L152 is based on the following experimental setup. We use the same instruction text and perform inference with two different input images. For each layer, we compute the cosine similarity between the instruction embeddings of these two image–text pairs; a lower similarity indicates stronger cross-modal interaction. As shown in Fig. 2, the cross-modality similarity in the shallow layers is close to 1, which means that even when the input image does not match the instruction, the instruction embeddings in shallow layers remain almost identical to those obtained with the correct image. This indicates that visual information has almost no cross-modal impact in the shallow layers.
>
> - > **Q3:What does p value mean in Figure 3?**
> - **A3**: To generate pruning schedules with different decay rates, we parameterize them using exponential decay (ED) and generalized exponential decay (GED) functions, without changing the rest of the framework. In the GED case, when the exponent parameter 0<𝑝<1, the schedule exhibits a faster initial drop than standard exponential decay, and smaller values of p lead to an even steeper early descent.
>
> > **W2: The analysis seems to be limited to a single type of language model, which lowers the credibility of the generality of the observed underlying vision-language fusion mechanism.**
>
> **A4**: In the paper, the goal of Section 2 is to derive insights from empirical observations so as to guide our model design, rather than to analyze a single language model in isolation. In fact, our experiments are not restricted to one LLM: as shown in Sec. 4.2 (Tab. 2), within the LLaVA framework we validate our method on two different LLM backbones, MobileLLaMA-2.7B and Vicuna-v1.5-7B (LLaVA-v1.5-7B). This cross-backbone consistency provides indirect evidence that the observed fusion mechanisms are not an artifact of a particular language model, but instead reflect a more general behavior.
>
> Regarding the observed vision–language fusion mechanism, we decompose the process into three stages: shallow layers acting as propagators and attention sinks, middle layers serving as the main vision–language fusion stage, and deep layers functioning as a language-dominant reasoning stage. This layered fusion pattern is obtained by identifying where the characteristic shallow and deep behaviors emerge. While the deep-layer behavior is largely consistent with prior observations in the literature, the propagation role of shallow layers is, to the best of our knowledge, a new finding specific to our analysis.
>
> Regarding the shallow-layer observation that the reviewers found compelling, we have added layer-wise representational dynamics analyses across different LLM backbones in Appendix F. These new experiments both demonstrate the generality of the effect and serve as the primary empirical basis for our shallow-layer observation.

---

> ### Author Response · Authors · 2025-12-03
> **Response to 8C26 (part 2)**
>
> > **W3: Under a similar training budget, it seems the proposed method achieves lower performance than PDrop. Though at a lower inference computation budget, the important metric is the inference latency, which the manuscript fail to provide. It is also confusing why the method, reducing so many visual tokens (64 compared to 270 in PDrop), requires similar training budget.**
>
> **A5**: We would like to clarify that Table 1 in our paper already reports a strictly matched budget setting. Under this directly comparable setup, HiDivDrop consistently outperforms PDrop (e.g., 98.4% vs. 96.8% at 86.1% pruning ratio and 98.3% vs. 94.2% at 88.9% pruning ratio). At even more aggressive compression, HiDivDrop still retains 96.5% of the baseline at 91.7% pruning ratio, whereas PDrop cannot reach this pruning level under the same budget.
>
> Although HiDivDrop reduces the effective number of visual tokens much more aggressively than PDrop, the actual training cost remains similar because the pruning pattern is dynamic and irregular, and current GPU kernels are still optimized for dense, fixed-shape attention. As a result, the underlying hardware cannot fully exploit the theoretical reduction in token count, which is a general limitation of token-pruning implementations rather than a drawback specific to HiDivDrop.
>
> To directly address the reviewer’s concern on efficiency, we also report end-to-end inference latency for HiDivDrop and PDrop under a common evaluation setting. As shown in the table below, HiDivDrop achieves better prefill latency than PDrop while attaining substantially higher visual-token compression.
>
> Specifically, for HiDivDrop we report three prefill times: (i) the actual latency measured with our current implementation, (ii) the latency after decoupling the vision-related KV projection operations from the main attention computation and running them in parallel, and (iii) the latency after reducing the number of dropping stages while maintaining a comparable overall compression ratio. Thanks to the late visual injection design, the KV projections for middle-layer visual tokens can be computed in advance and parallelized with shallow language layers, which shortens the critical-path prefill time.
>
> **Table R1. Efficiency comparison across three LLM backbones within the LLaVA-1.5 framework.**
> | Model | Method | Avg. Vis. Tokens | Train hours | Infer TFlops | Prefill Time (ms) | MME-P | MMB | GQA | VizWiz | VQA-T | Avg (%) |
> |-|-|-|-|-|-|-|-|-|-|-|-|
> | LLaVA-1.5-MobileLLaMA-2.7B | Vanilla | 576 | 108.4 | 1.52 | 35.3 | 1258.2 | 57.0 | 59.4 | 32.6 | 48.6 | 100.0 |
> |                            | PDrop   | 270 | 50.3 | 0.70 | 28.7 | 1231.1 | 54.3 | 57.0 | 30.9 | 47.5 | 96.3 |
> |                            | ours    | 64 ↓ 206 | 45.6 | 0.17 | 25.4/25.1/22.0| 1206.6 | 53.1 | 56.1 | 30.4 | 47.2 | 94.8 ↓ 1.5 |
> | LLaVA-1.5-Vicuna-1.5-7B | Vanilla | 576 | 159.3 | 3.82 | 63.6 | 1506.5 | 64.7 | 61.9 | 50.1 | 58.2 | 100.0 |
> |                         | PDrop   | 270 | 107.3 | 1.78 | 43.7 | 1490.1 | 63.9 | 61.7 | 52.4 | 57.7 | 100.2 |
> |                         | ours    | 64 ↓ 206 | 94.4 | 0.42 | 32.6/31.8/28.8 | 1474.3 | 63.2 | 60.5 | 52.6 | 55.2 | 98.6 ↓ 1.6 |
> | LLaVA-1.5-Vicuna-1.5-13B | Vanilla | 576 | 297.2 | 7.44 | 122.8 | 1529.9 | 68.5 | 63.5 | 53.6 | 61.2 | 100.0 |
> |                          | PDrop   | 270 | 213.7 | 3.47 | 74.9 | 1555.2 | 68.8 | 63.1 | 53.7 | 60.8 | 100.3 |
> |                          | ours    | 64 ↓ 206 | 175.8 | 0.82 | 48.6/46.6/43.5| 1497.2 | 66.9 | 62.1 | 56.3 | 58.0 | 98.7 ↓ 1.6 |

---

> ### Author Response · Authors · 2025-12-03
> **Response to 8C26 (part 3)**
>
> > **W4: The experimental analysis on the proposed method is not sufficient. For example, it would be crucial to know how the the training/inference cost and performances change when the injection becomes later (starting from first-layer injection).**
>
> **A6**: We thank the reviewer for pointing out this missing analysis and agree that it is important to understand how the training/inference cost and performance change when the injection layer varies (including the case of first-layer injection). As partially shown in Fig. 7 and Tab. 7, we have already evaluated the performance of late injection at different layers (5th–11th). Building on this, we will extend the analysis by additionally reporting the changes in training and inference cost when shifting the injection point from the first layer to these later layers, including FLOPs, wall-clock latency, and task performance, and will include the results in the revised version.
>
> **Table R2. Ablation study across visual perception layers for Late Injection.**
> | Model                         | Train Hours | Infer Flops | Prefill Time (ms)| MME-P  | MMB  | MMB-CN | GQA  | VQAv2 | SQA-IMG | VizWiz(val) | TextVQA | POPE | SEED-IMG | MMStar | Avg (%) |
> |-|-|-|-|-|-|-|-|-|-|-|-|-|-|-|-|
> | Base Model (LLaVA-v1.5-7B)    | 159.3 | 3.82 | 63.6 | 1506.5 | 64.7 | 58.1 | 61.9 | 78.5 | 69.5 | 54.4 | 58.2 | 86.8 | 66.2 | 33.7 | 100.0 |
> | E2                            | 125.9 | 3.70 | 62.7 | 1442.1 | 67.6 | 59.7 | 63.6 | 78.8 | 68.3 | 53.6 | 58.1 | 87.5 | 57.1 | 33.3 | 100.3 |
> | E3                            | 124.1 | 3.58 | 61.8 | 1499.3 | 67.4 | 59.3 | 63.1 | 78.9 | 69.6 | 51.1 | 58.6 | 87.3 | 66.6 | 34.5 | 100.6 |
> | E4                            | 121.7 | 3.46 | 61.0 | 1498.1 | 66.4 | 59.5 | 63.2 | 78.8 | 70.3 | 53.3 | 58.5 | 87.3 | 66.1 | 34.5 | 100.8 |
> | E5                            | 120.7 | 3.34 | 60.1 | 1441.9 | 66.0 | 59.9 | 62.4 | 78.5 | 69.5 | 55.7 | 55.9 | 86.5 | 65.6 | 34.0 | 100.0 |
> | E6                            | 118.7 | 3.32 | 59.2 | 1442.5 | 65.8 | 58.3 | 62.4 | 78.5 | 69.3 | 56.1 | 56.5 | 86.8 | 66.1 | 33.3 | 99.8  |
> | E7                            | 118.0 | 3.10 | 58.4 | 1413.8 | 66.2 | 58.8 | 62.2 | 78.3 | 69.8 | 53.0 | 56.6 | 86.4 | 65.3 | 33.1 | 99.0  |
> | E8                            | 116.5 | 2.98 | 57.5 | 1424.1 | 65.1 | 58.2 | 62.7 | 78.3 | 69.1 | 54.1 | 57.3 | 87.1 | 65.9 | 31.9 | 98.9  |
> | E9                            | 116.1 | 2.86 | 56.6 | 1444.4 | 65.4 | 57.9 | 61.5 | 77.9 | 68.9 | 56.9 | 56.1 | 86.5 | 65.3 | 32.7 | 99.2  |
>
> > **W5: It seems costly to determine the injection layer and the exit layer for each given MLLM, how can the hyperparameter be efficiently determined?**
>
> **A7**: We agree that a naive grid search over all possible injection and exit layers would be prohibitively expensive. In practice, however, HiDivDrop uses a much more efficient two-stage procedure. First, we apply a training-free diagnostic as in Sec. 2 to identify a narrow window of roughly 3 layers. Second, we only train a few configurations within this window to select the final injection and exit layers, rather than searching over all layers. This procedure does not require the full training set: for example, we can use a smaller subset such as LLaVA-ICONS-133K instead of the full LLaVA-Mix-665K for validation, which keeps the additional cost modest compared to training the base MLLM.
>
> Moreover, we observe that even without exhaustive tuning, different choices of injection/exit layers inside the identified window already yield very similar performance (e.g., I7 99.0%, I8 99.1%, I9 99.3%, I9E24 98.7% I9E25 99.2%, I9E26 100.0% performance, see Appendix F Tab.7, indicating that HiDivDrop is reasonably robust to these hyperparameters.

---

### Official Review · Reviewer_gcbi · 2025-11-10

**Soundness:** 2
**Presentation:** 3
**Contribution:** 2
**Rating:** 4
**Confidence:** 4

**Summary:**

This paper studies the visual token reduction problem for the multimodal large language models (MLLMs). The author analyzes the diverse impact of the visual tokens in shallow, middle, and deep layers for LLaVA-1.5-3B and LLaVA-1.5-7B, and proposes a hierarchical division-based vision token dropping (HiDivDrop) method where the shallow layers are handled with Late Injection and the deep layers are handled with Early Exit, and apply the Concave Pyramid Dropping in the middle layers to progressively reduce vision tokens. Experiments are conducted on the LlaVA-1.5 architecture with different LLM backbones and 11 mainstream benchmarks, verifying the effectiveness of the proposed method.

**Strengths:**

1. The proposed method is effective in reducing a large amount of visual tokens compared to the state-of-the-art methods on extensive benchmarks.

**Weaknesses:**

1. Generalizability of the analysis in Section 2. The analysis in Section 2 serves as the primary motivation for the proposed method. However, the details about the training and evaluation datasets, training configuration, and the exact model type (e.g., the LLM backbones) are missing. It is challenging to convince the reviewer that the empirical observation in Section 2 is generally applicable to various pretrained datasets, evaluation tasks, training configurations, and model types for MLLM. The author should provide more context for the empirical observation to let the reader understand the generalizability of the observation.

2. The applicability of the Joint Visual Layer Reduction. In Section 3.1, lines 227-228 and lines 232-234, the author selects the specific layers for Late Vision Injection and Early Vision Exit based on the empirical observations in Section 2. However, as mentioned in Weakness 1, the pretrained datasets, evaluation tasks, training configurations, and model types for MLLM are not clarified in the main paper, making it hard for the reviewer to understand whether this layer selection is also applicable when we change to other pretraining datasets, evaluation tasks, and model types.

Moreover, the author does not provide a concrete description of how to identify those in practice; for example, should we evaluate the MLLM performance, as in Section 2, every time we are given new datasets, evaluation tasks, and model types? Should we analyze the training dataset or the validation dataset to identify the optimal layers? How can we guarantee that the selection is optimal when we change the model training recipe?

3. The choice of MLLM architecture. In Section 4, the author only evaluates the LLaVA framework as the MLLM. However, it is not clear whether the proposed method can be applied to other MLLM frameworks.

4. Technical Contribution is limited. Most of the methods used in Section 3 have existed previously, and the overall novelty of the proposed method is limited.

**Questions:**

Please refer to the Weaknesses section for details.

---

> ### Author Response · Authors · 2025-12-03
> **Response to gcbi (part 1)**
>
> We thank you for your thorough review and appreciate both the positive feedback and the constructive comments. Please find our responses to your points below. In the revised manuscript, newly added or substantially modified text is highlighted in blue, while previously existing passages that already address your concerns are highlighted in yellow for ease of reference. Please let us know if you have any further questions or suggestions.
>
> > **W1: Generalizability of the analysis in Section 2. The analysis in Section 2 serves as the primary motivation for the proposed method. However, the details about the training and evaluation datasets, training configuration, and the exact model type (e.g., the LLM backbones) are missing. It is challenging to convince the reviewer that the empirical observation in Section 2 is generally applicable to various pretrained datasets, evaluation tasks, training configurations, and model types for MLLM. The author should provide more context for the empirical observation to let the reader understand the generalizability of the observation.**
>
> **A1**: The analysis in Sec.2 includes 4 experiments: (a) intra-model refinement (Shallow Layers), (b) cross-modal influence (Shallow Layers), (c) sparsity of vision tokens (Middle Layers), and (d) early exit (Deep Layers).
>
> - As noted in the Global Response, our observation is conducted within the LLaVA architecture, a widely used MLLM framework in which the LLM backbone can be replaced. In the original manuscript, Experiments (a) and (b) report results on LLaVA-v1.5-MobileLLaMA-2.7B and LLaVA-v1.5-7B, while Experiments (c) and (d) are based on LLaVA-v1.5-7B. Further details about the LLM backbones are provided in Appendix C. We now further extend the layer-wise representational dynamics analysis to additional LLM backbones in Appendix F. These new results show that the shallow-layer invariance and stratification phenomena consistently appear across different backbones, which supports the generality of the observations beyond a single LLM.
>
> - Regarding the training setup: Experiments (a), (b), and (d) are training-free analyses based on existing model checkpoints, where we do not retrain the model to adapt to Late Injection or Early Exit, but instead use them to probe the inherent behavior of the architecture. By contrast, Experiment (c) follows the standard LLaVA training protocol and is used to validate the concave pyramid pruning scheme under full training.
>
> - In Sec. 2, we intentionally use GQA as a running example to introduce and visualize the phenomena, as it is a representative benchmark for real-world visual reasoning, and we observe similar trends on other evaluation tasks as well. More importantly, in our main experiments we systematically evaluate the proposed method on 11 diverse benchmarks and consistently observe the same layer-wise trends and performance behavior. These results, summarized in Sec. 4 and the corresponding tables, provide strong evidence that the empirical observations in Sec. 2 are not artifacts of a single dataset or task, but reliably hold across a broad spectrum of evaluation settings.

---

> > ### Author Response · Authors · 2025-12-03
> > **Response to gcbi (part 2)**
> >
> > >**W2:The applicability of the Joint Visual Layer Reduction. In Section 3.1, lines 227-228 and lines 232-234, the author selects the specific layers for Late Vision Injection and Early Vision Exit based on the empirical observations in Section 2. However, as mentioned in Weakness 1, the pretrained datasets, evaluation tasks, training configurations, and model types for MLLM are not clarified in the main paper, making it hard for the reviewer to understand whether this layer selection is also applicable when we change to other pretraining datasets, evaluation tasks, and model types.**
> >
> > **A2**: We thank the reviewers for pointing out that the empirical analysis in Section 2 lacked some background information, which we have addressed in the revised version.
> >
> > We clarify that the analysis in Sec. 2 is intended as a motivational diagnostic study to identify representative shallow / middle / deep regions, while the full experimental setup of HiDivDrop (models, pretraining data, training configuration, and benchmarks) is specified in Sec. 4.1, Appendixes C and D. Although we use GQA as the guiding dataset for selecting the injection and exit layers, this layer selection strategy generalizes well to a broader set of evaluation benchmarks, as demonstrated in Sec. 4.3 Late Injection and Early Exit.
> >
> > In Sec. 4.2 (Tab. 1), we conduct controlled-budget experimentsat three different compression ratios, where under each fixed computation budget the HiDivDrop configuration is kept identical across all datasets. Across 11 diverse benchmarks and all three compression ratios, HiDivDrop achieves average accuracies of 98.4%, 98.3%, and 96.5 at the 86.1%, 88.9%, and 91.7% pruning settings, respectively, outperforming the strongest state-of-the-art baseline by +0.5%, +3.0%, and +4.9, demonstrating that the method is not tied to a single dataset or task.
> >
> > Furthermore, we instantiate HiDivDrop with multiple LLM backbones within the LLaVA framework (e.g., MobileLLaMA-2.7B and 7B-scale models), as shown in Sec.4.2 (Tab.2), and observe highly consistent performance behavior. This indicates that the proposed strategy remains applicable across different LLM models.

---

> > > ### Author Response · Authors · 2025-12-03
> > > **Response to gcbi (part 3)**
> > >
> > > > **W: Moreover, the author does not provide a concrete description of how to identify those in practice; for example, should we evaluate the MLLM performance, as in Section 2, every time we are given new datasets, evaluation tasks, and model types? Should we analyze the training dataset or the validation dataset to identify the optimal layers? How can we guarantee that the selection is optimal when we change the model training recipe?**
> > >
> > > **A3**: We thank the reviewer for raising this practical question. Our goal is not to require an exhaustive multi-layer evaluation on every new dataset and task, but to offer a lightweight diagnostic procedure that estimates the effective shallow, middle and deep layers once for a given backbone. Specifically, we proceed in two training-free steps. First, on a representative multimodal QA benchmark, we perform a layer-wise probe of the backbone to be compressed: we compute the cosine similarity of single-modality representations (e.g., visual features) between adjacent layers, and take a pronounced local maximum of the visual-similarity curve as the center of the late-injection window. Second, we conduct training-free, layer-wise early-exit experiments on the same benchmark, and select the earliest layer whose performance is nearly indistinguishable from the full model as the right boundary of the early-exit window. This analysis provides a one-shot coarse estimate of the injection and exit layers.
> > >
> > > Importantly, the objective of this work is not to pinpoint a unique “optimal” pair of late-injection and early-exit layers. Instead, we aim to uncover key mechanisms of visual information processing in MLLMs, thereby providing insights for future research and inspiring architecture design. As shown in Fig. 7, even when the injection/exit layers deviate from the specific choices in our main setting, the model still retains the vast majority of its performance. In practice, the exact layer indices can therefore be left to the user and adjusted flexibly according to the desired performance–efficiency trade-off.
> > >
> > > > **W3: The choice of MLLM architecture. In Section 4, the author only evaluates the LLaVA framework as the MLLM. However, it is not clear whether the proposed method can be applied to other MLLM frameworks.**
> > >
> > > **A4**: Regarding the choice of MLLM architecture, please refer to the Global Response, “Experiment Settings (including the choice of MLLM architecture and training strategy)”, where we explain why we focus on LLaVA-1.5 and how the proposed method generalizes across different LLM backbones within this framework.
> > >
> > > Importantly, within the LLaVA architecture our method is not tied to a specific language backbone: in the current manuscript we already evaluate on two distinct LLM backbones, Vicuna-7B-v1.5 and MobileLLaMA-2.7B, and observe consistent gains.

---

> ### Author Response · Authors · 2025-12-03
> **Response to gcbi (part 4)**
>
> > **W4: Technical Contribution is limited. Most of the methods used in Section 3 have existed previously, and the overall novelty of the proposed method is limited.**
>
> **A5**: We respectfully disagree with the assessment that the technical contribution and overall novelty of our method are limited. Although our work leverages known building blocks such as progressive token pruning and early exit, Section 3 does not simply reuse existing techniques; instead, HiDivDrop introduces a unified, hierarchy-aware framework that (i) diagnoses redundancy across depth and token types, and (ii) designs coordinated late injection, multi-stage dropping, and early exit based on these findings. Below, we clarify the core innovations of our approach and explain in detail how they go beyond prior methods.
>
> - Reinterpreting the role of shallow layers: We identify a common misconception in the community regarding shallow layers. Prior work typically assumes that they are crucial for cross-modal fusion [1,2], whereas our empirical analysis shows that they mainly act as simple propagation layers with limited multimodal interaction. Guided by this finding, we introduce a late injection strategy for visual tokens, which, to the best of our knowledge, has not been explored in prior compression work on MLLMs.
>
> - Concave pyramid pruning schedule + pruning layer selection: Existing pruning approaches typically adopt either one-shot pruning [3,4] or progressive schemes with uniform pruning ratios and evenly spaced pruning layers [1]. We depart from this paradigm in two key aspects. First, we introduce a concave pyramid pruning schedule that prunes visual tokens more aggressively in early fusion stages and more conservatively in deeper layers, enabling rapid redundancy removal while preserving high-level semantics. Second, we propose to determine pruning layers via inter-layer attention similarity, rather than placing them uniformly in depth, so that pruning is aligned with where fusion dynamics actually change. To the best of our knowledge, this joint design of concave pyramid pruning across depth and attention-guided pruning layer selection does not exist in prior visual token compression work on MLLMs
>
> - A simple yet principled HiDivDrop design with practical deployability: Building on the above insights, HiDivDrop is conceptually simple yet structurally well-motivated. We skip visual tokens in shallow and deep layers and apply a differentiable top-k selection mechanism to visual tokens in the middle layers, enabling substantially higher visual-token compression while largely preserving performance. In addition, we use persistent positional identifiers for dynamically pruned visual tokens and provide an implementation that is fully compatible with FlashAttention-style kernels, ensuring stability and efficiency in both training and inference. To the best of our knowledge, this position-consistent dynamic pruning implementation has not been addressed in prior MLLM token compression methods.
>
> [1] Xing L, Huang Q, Dong X, et al. Pyramiddrop: Accelerating your large vision-language models via pyramid visual redundancy reduction[J]. arXiv preprint arXiv:2410.17247, 2024.
>
> [2] Zhang S, Fang Q, Yang Z, et al. Llava-mini: Efficient image and video large multimodal models with one vision token[J]. arXiv preprint arXiv:2501.03895, 2025.
>
> [3] Chen L, Zhao H, Liu T, et al. An image is worth 1/2 tokens after layer 2: Plug-and-play inference acceleration for large vision-language models[C]//European Conference on Computer Vision. Cham: Springer Nature Switzerland, 2024: 19-35.
>
> [4] Shao Z, Wang M, Yu Z, et al. Growing a twig to accelerate large vision-language models[J]. arXiv preprint arXiv:2503.14075, 2025.

---

### Comment · Area_Chair_T98i · 2025-11-23
**Reviewer & Author Discussion**

Hi Reviewers,

Please kinly and actively participate in the review-author dicussion, raise your further concerns so that the authors can explain more, and make your final decisions.

---

### Author Response · Authors · 2025-12-03
**Global Response (part 1)**

We sincerely thank all the reviewers for their thorough reviews and valuable feedback. We are glad to hear that we provide a systematic layer-wise analysis that clarifies where vision–language fusion truly occurs and offers useful guidance for future VLM research (8C26, VaQT), especially the novel insight that shallow layers mainly act as propagators with limited fusion capacity, so that shallow vision tokens are dispensable and can be safely bypassed (VaQT). Building on these insights, the resulting model is regarded as simple yet reasonable (8C26, VaQT), and achieves strong performance even under highly compressed visual token budgets (all reviews).

Across the reviews, we identified a few shared themes, which we address below:

- **Experiment Settings (including the choice of MLLM architecture and training strategy)**: (Reviewer: gcbi, VaQT)
    - LLaVA remains one of the most widely used and actively studied open-source MLLM architectures in the academic community, making it a natural and representative testbed for analyzing vision-token compression [1-3].
    -  LLaVA fully open-sources both its training data and training pipeline, which is crucial for our work. We focus on integrating the compression mechanism into the model design itself, so that we can explicitly exploit the internal vision–language fusion dynamics to identify and remove redundant visual tokens, rather than treating compression as a purely post-hoc add-on,  and our method is compatible with both the pre-training and SFT stages. (Reviewer: VaQT)
    - In contrast, for many other popular open-source VLMs (e.g., Qwen2.5-VL, Gemma3-VL, Qwen3-VL), only the checkpoints are available while the full training dataset is not, which restricts us to purely post-hoc compression without the kind of training-time intervention we perform here. We view such checkpoint-based, post-training compression as an equally important and complementary direction, and we plan to investigate this setting in future work.
    - To rule out the concern that our findings might be an artifact of under-training or a particular training recipe, we also conduct incremental training experiments by continuing to train both the original LLaVA-1.5 baseline and the HiDivDrop-equipped model on additional multimodal instruction data (≈335k extra samples from the Open-LLaVA-Next recipe). As summarized in Table R3 of the rebuttal, both models benefit from the extra training, and HiDivDrop consistently retains its advantage, indicating that our conclusions are not tied to a fragile or under-trained setting.  (Reviewer: VaQT)

- **The generalization of the insight and proposed method:**
    - Within the LLaVA architecture, we are able to evaluate different LLM backbones. For example, in the original manuscript we evaluate both Vicuna-7B-v1.5 and MobileLLaMA-2.7B, which represent two distinct language backbones. In the rebuttal, we further add experiments on a larger backbone, Vicuna-13B-v1.5. As shown in Table R2, we observe the same patterns and consistent performance from HiDivDrop. These additional results further support the generality of our findings across model sizes and backbones.(Reviewer: doEa)
    - In the same Table R2, we also report prefill latency for both HiDivDrop and PDrop under matched settings, showing that HiDivDrop attains better latency while achieving substantially higher visual-token compression. Notably, thanks to our late visual injection strategy, the KV projection operations for visual tokens can be decoupled from the main attention path and computed earlier in shallow layers in parallel with the full text-token computation, effectively reducing the latency of this component to zero. We also report this decoupled variant in Table R2. (Reviewer: 8C26)
    - Regarding the shallow-layer observation that the reviewers found compelling, we have added layer-wise representational dynamics analyses across different LLM backbones and different framework in Appendix F, including LLaVA-v1.5-MobileLLaMA-2.7B, LLaVA-v1.5-Vicuna-7B, LLaVA-v1.5-LLaMA3.1-8B, LLaVA-v1.5-Vicuna-13B, LLaVA-NeXT-Vicuna-7B and LLaVA-NeXT-LLaMA3-8B. These new experiments both demonstrate the generality of the effect and serve as the primary empirical basis for our shallow-layer observation. (Reviewer: 8C26, gcbi)


[1] Xing L, Huang Q, Dong X, et al. Pyramiddrop: Accelerating your large vision-language models via pyramid visual redundancy reduction[J]. arXiv preprint arXiv:2410.17247, 2024.

[2] Shao Z, Wang M, Yu Z, et al. Growing a twig to accelerate large vision-language models[J]. arXiv preprint arXiv:2503.14075, 2025.

[3] Tang H, Shen C. Learning Compact Vision Tokens for Efficient Large Multimodal Models[J]. arXiv preprint arXiv:2506.07138, 2025.

---

> ### Author Response · Authors · 2025-12-03
> **Global Response (part 2)**
>
> - **Consistency of the compression strategy across tasks:**
>     - Our training strictly follows the standard LLaVA-1.5 pipeline, i.e., we train on the same pre-training and instruction-tuning datasets as LLaVA and evaluate on the same diverse set of benchmarks. In other words, we learn a single, task-agnostic compression strategy that is shared across all tasks, without any task-specific tuning of the pruning policy. (Reviewer: VaQT)
>     - Task-specific or even sample-specific compression is a potentially more powerful but also substantially harder topic, and falls outside the scope of this work. We view it as an important topic for future research and plan to explore it in follow-up studies.
>
> - **Technical contribution and novelty (summary)**:  Beyond empirical observations, HiDivDrop makes several conceptual and methodological contributions to visual token compression in MLLM
>     -  First, our layer-wise analysis overturns the common assumption that shallow layers are the main locus of cross-modal fusion and instead reveals that they mainly act as propagators, which directly motivates our late visual injection design.
>     - Second, we introduce a hierarchy-aware concave pyramid pruning schedule together with visual attention similarity–guided pruning layer selection, rather than using uniform ratios and evenly spaced pruning layers as in prior work.
>     - Finally, we show that these design choices can be realized in a position-consistent implementation that remains compatible with FlashAttention-style kernels, enabling practical deployment during both training and inference.
>     - Taken together, these components form a framework that tailors token pruning to the true hierarchical roles of MLLM layers, which, to the best of our knowledge, has not been explored in prior MLLM token compression work.

---

### Meta-Review · Area_Chair_tNpn · 2025-12-20

**Summary:**

Based on the discussion, the reviewers' primary concerns revolved around four main themes:

- Generality and Robustness of the Core Insight (Section 2): Reviewers gcbi and 8C26 questioned whether the novel observation—that shallow layers act as propagators with limited fusion—is generalizable. They raised concerns about it being specific to the LLaVA-1.5 architecture, a particular training recipe, or a single dataset (GQA), and asked for validation across more backbones, datasets, and training scales.

- Practical Applicability and Hyperparameter Sensitivity: Reviewers gcbi and 8C26 expressed concerns about the practical cost of determining the optimal late-injection and early-exit layers for new models/datasets. They questioned if the method required an expensive, task-specific search for these hyperparameters.

- Technical Contribution and Novelty: Reviewer gcbi felt the technical contribution was limited, arguing that the method's components (progressive pruning, early exit) are not novel by themselves. This point touches on whether the paper's primary contribution is the empirical insight or the method built upon it.

- Experimental Completeness and Clarity: Reviewer 8C26 noted the lack of reported inference latency, a crucial metric for an efficiency-focused paper, and was confused about training cost vs. token reduction. Reviewer VaQT questioned the focus on the "older" LLaVA-1.5 model and whether conclusions hold for newer, more powerful VLMs. Reviewers doEa and 8C26 found the writing and structure (especially Section 2's purpose and the placement of tables) unclear, hindering readability.

**Reviewer Concerns:**

- Technical Contribution/Novelty: The authors' defense (A5 to gcbi) reframed the contribution as a unified, hierarchy-aware framework built on a new empirical insight. They highlighted the novel combination of late injection (inspired by the insight), concave pyramid pruning, and attention-guided layer selection. While this is compelling, reviewer gcbi's original score (4) suggests they may still view the synthesis as incremental. This is likely the most subjective and unresolved point.

- Scope to Newer VLMs (Qwen, Gemma): The authors provided a reasonable justification for using LLaVA (open data/pipeline, representative testbed) and showed generalization across backbones within that architecture. However, the rebuttal does not extend results to fundamentally different architectures like Qwen2.5-VL. Their plan to explore this in future work is noted, but it remains an outstanding limitation of the current work's scope.

**Reviewer Scores:**

- Reviewer gcbi (Initial: 4): Their main concerns (generality, hyperparameter search) were addressed with new experiments and clear methodology.

- Reviewer 8C26 (Initial: 4): Their concerns about generality (addressed with new backbones), missing latency data (provided), and hyperparameter cost (clarified) were all met effectively. The request for more ablation on injection timing was also addressed.

- Reviewer VaQT (Initial: 6): Their concerns about justification for shallow layers (expanded), undertraining (addressed with incremental training), and training strategy clarity (clarified) were well-answered. The scope limitation (newer models) remains but is mitigated.

- Reviewer doEa (Initial: 6): Their requests for more backbones (added Vicuna-13B) and analysis of knowledge drift (provided with new metrics) were fulfilled.

---

### Decision · Program_Chairs · 2026-01-26

Accept (Poster)